# Photocatalytic toluene oxidation with nickel-mediated cascaded active units over Ni/Bi$_2$WO$_6$ monolayers

Yingzhang Shi [1,2,4], Peng Li[1,4], Huiling Chen[1], Zhiwen Wang[1,2], Yujie Song[1,2], Yu Tang[1], Sen Lin [1] ✉, Zhiyang Yu [1] ✉, Ling Wu [1] ✉, Jimmy C. Yu [1,3] & Xianzhi Fu[1]

Adsorption and activation of C–H bonds by photocatalysts are crucial for the efficient conversion of C–H bonds to produce high-value chemicals. Nevertheless, the delivery of surface-active oxygen species for C–H bond oxygenation inevitably needs to overcome obstacles due to the separated active centers, which suppresses the catalytic efficiency. Herein, Ni dopants are introduced into a monolayer Bi$_2$WO$_6$ to create cascaded active units consisting of unsaturated W atoms and Bi/O frustrated Lewis pairs. Experimental characterizations and density functional theory calculations reveal that these special sites can establish an efficient and controllable C–H bond oxidation process. The activated oxygen species on unsaturated W are readily transferred to the Bi/O sites for C–H bond oxygenation. The catalyst with a Ni mass fraction of 1.8% exhibits excellent toluene conversion rates and high selectivity towards benzaldehyde. This study presents a fascinating strategy for toluene oxidation through the design of efficient cascaded active units.

The selective activation and oxidation of C–H bonds to produce valuable oxygenated products has always been a research hotspot in catalysis[1,2]. However, this process remains challenging, particularly under mild conditions, due to the difficult adsorption of C–H bonds and the high bond dissociation energy of 70–130 kcal mol$^{-1}$ [3,4]. Heterogeneous photocatalysis offers a promising approach for the activation and oxidation of C–H bonds under mild conditions. However, addressing the low photocatalytic efficiency and unclear surface reaction mechanism for C–H bond oxidation remains a challenge. Although many strategies have been developed to improve the photocatalytic efficiency of semiconductors based on classical energy band theory, such as constructing heterojunctions, doping with other elements, and loading precious metals[5–7], less attention has been paid to the surface/interfacial chemical processes of C–H bond oxidation. During photocatalytic C–H bond oxidation, three main aspects should be considered: (i) activating C–H bonds and O$_2$ molecules, (ii) separating and migrating photogenerated carriers, and (iii) surface redox

reaction and transfer of active O species[8–13]. It is crucial to comprehensively optimize these processes to achieve efficient photocatalytic performance.

Designing potential active sites on a photocatalyst is necessary to enhance the activation of C–H bonds. Recent studies suggest that the strongly polarized environment favors the activation of C–H bonds[14]. However, the polarized sites consisting of the neighboring lattice atoms, such as the classical Lewis acid-base pairs (CLPs), are generally difficult to adsorb and activate the C–H bonds[15,16]. Frustrated Lewis pairs (FLPs) formed by sterically hindered Lewis acid (LA) and Lewis base (LB) pairs have been reported to efficiently activate substrate molecules, including typical reactions such as amidation, hydrogen splitting, and CO$_2$ reduction[17–19]. As a result, FLPs are suitable candidates for the C–H bond activation due to their stronger polarization environment and frustrated reaction centers compared to CLPs.

The activation of O$_2$ molecules is also crucial for the oxidation of C–H bonds. Activated O$_2$ molecules are more effective in the

[1]State Key Laboratory of Photocatalysis on Energy and Environment, Fuzhou University, Fuzhou, Fujian 350116, China. [2]School of Chemistry and Chemical Engineering, Hainan University, Haikou, Hainan 570228, China. [3]Department of Chemistry, The Chinese University of Hong Kong, Hong Kong, China. [4]These authors contributed equally: Yingzhang Shi, Peng Li. ✉e-mail: slin@fzu.edu.cn; yuzyemlab@fzu.edu.cn; wuling@fzu.edu.cn

oxygenation of C−H bonds compared to free $O_2$, as they can be converted to surface-active oxygen species[11–13]. Defect engineering presents a good opportunity for $O_2$ activation. Surface defects may cause the formation of coordinatively unsaturated sites (CUSs), which typically have the ability to adsorb and coordinatively activate $O_2$ molecules, facilitating the generation of active O species[20,21]. However, due to the discrete catalytic active centers, the transfer of surface active oxygen species from their original active sites to other active sites for the oxygenation of C−H bond usually requires overcoming resistance, which significantly hinders the catalytic efficiency. Therefore, the concept of constructing cascaded active units (CAUs) that include FLPs and CUSs is both innovative and feasible. The CAUs are active units that allows FLP and CUS to be connected in series, being spatially interconnected or in close proximity to each other. This approach can not only regulate the activation of C−H bonds and $O_2$ but also mitigate the inhibitory effect of active O transfer on photocatalytic performance due to the very close distance between the two active sites. The rational construction of CAUs poses an extremely challenging problem.

Herein, we developed a series of Ni-doped monolayer $Bi_2WO_6$ nanosheets (Ni/BWO) with varying Ni mass fractions for the photocatalytic selective oxidation of toluene. Particularly, Ni/BWO with a Ni mass fraction of 1.8% shows a toluene conversion rate as high as $4560\ \mu mol\ g^{-1}\ h^{-1}$ and a high selectivity towards benzaldehyde. Structural characterizations and density functional theory (DFT) studies confirm that the doping of Ni on monolayer $Bi_2WO_6$ nanosheets can induce so-called cascaded active units (CAUs). These CAUs include surface oxygen vacancies ($O_Vs$), unsaturated W atoms, and FLPs sites composed of spatially separated Bi and O atoms, which are well connected through the mediation of Ni dopants. The combined experimental and DFT calculation results indicate that CAUs can be prepared rationally through in-situ lattice atom substitution by metal dopants and have a remarkable synergistic effect for the selective oxidation of C−H bonds.

## Result and discussion
### Characterization of the prepared samples
A series of Ni/BWO samples with different Ni mass fractions were prepared via a modified one-step solvothermal route[22]. The schematic diagram is shown in Fig. 1a. The actual mass fractions of Ni were determined by Inductively Coupled Plasma Mass Spectrometry (ICP-MS) to be 0.9%, 1.8%, and 2.6% (Supplementary Table 1). The corresponding samples were named as 0.9 Ni/BWO, 1.8 Ni/BWO, and 2.6 Ni/BWO, respectively. The X-ray diffraction (XRD) patterns (Fig. 1b (2θ from 25° to 35°) and Supplementary Fig. 1 (2θ from 5° to 70°)) show that all the samples are well matched with orthorhombic $Bi_2WO_6$ (JCPDS No. 73-2020; lattice parameters: a = 5.457 Å, b = 5.436 Å, and c = 16.427 Å)[4,23]. Interestingly, the XRD peaks shift to higher 2-theta with Ni doping, suggesting the contracted lattice of BWO, i.e., the substitution of Bi or W atoms by Ni atoms[24]. Fourier transform infrared (FT-IR) spectra (Fig. 1c) provide further microstructural details of these samples. The characteristic peaks of the O-W-O and W-O-W stretching vibrations at 736 and 577 $cm^{-1}$ are gradually weakened with the introduction of Ni[25], especially the O-W-O stretching peaks shift to low wavenumbers, indicating that some of the W atoms are substituted by Ni atoms. Two formed peaks at 955 and 665 $cm^{-1}$ are attributed to the short W=O and Ni-O stretching vibrations[25,26], respectively, suggesting that the ordered crystal structure of BWO is disrupted and part of the lattice oxygen atoms are lost, yielding numerous coordinatively unsaturated W atoms. Raman spectra (Fig. 1d) also show a characteristic peak at 782 $cm^{-1}$ assigned to the antisymmetric $A_g$ mode of O-W-O.[4,27] The vibrational intensity becomes weaker with Ni doping, also indicating the substitution of W atoms by Ni atoms. Consistent with the FTIR results, the symmetric stretching peak of the short terminal W = O at 955 $cm^{-1}$ is observed after the introduction of Ni[4,27,28],

illustrating that Ni breaks the ordered crystal arrangement of BWO accompanied by the formation of the $O_V$ and coordinatively unsaturated W atoms. $O_Vs$ have a spatial separation effect favoring the formation of FLPs[14,16].

The morphology and structure of the as-prepared samples were studied by field-emission scanning electron microscopy (SEM), transmission electron microscopy (TEM), and atomic force microscopy (AFM). Supplementary Fig. 2 illustrates that both 1.8 Ni/BWO and BWO samples exhibit similar nanosheet morphology, with an average thickness of approximately 1.5 nm (Fig. 1e and Supplementary Fig. 3). This measured thickness closely aligns with the interlayer spacing along the c-axis of orthorhombic $Bi_2WO_6$[29], indicating the successful synthesis of monolayer nanosheets, independent of the introduction of Ni atoms. These ultrathin nanosheets are well-resolved in subsequent low-magnified TEM images (Supplementary Fig. 4a: BWO, Fig. 1f: 1.8 Ni/BWO) and show a lattice spacing of 0.27 nm corresponding to the {002} planes of $Bi_2WO_6$[4,29,30] Notably, in contrast to the long-range ordered structure of the BWO sample (Supplementary Fig. 4b), lattice-disordered regions (highlighted by dotted lines) are observed within the 1.8 Ni/BWO counterpart (Fig. 1g). This observation suggests that the incorporation of Ni elements introduces a high density of surface defects within the $Bi_2WO_6$ nanosheets, which is potentially beneficial for the formation of FLPs[31] The specific surface areas of the 1.8 Ni/BWO and BWO nanosheets are relatively close (Supplementary Fig. 5). These results demonstrate that the introduction of Ni atoms leads to the emergence of surface defects in the Ni/BWO sample, with only a minor effect on the morphology. In the absence of Ni, $Bi_2WO_6$ with good crystallinity is formed by the combination of $[BiO]^+$-$[WO_4]^{2-}$-$[BiO]^+$ structural units[29,32] resulting in saturation of the coordination of most surface metal atoms in BWO. These CLPs, based on adjacent lattice atoms of $Bi_2WO_6$, are generally difficult to activate C−H bonds. Upon the introduction of Ni, the long-range ordered $[BiO]^+$-$[WO_4]^{2-}$-$[BiO]^+$ structural units are disrupted, leading to the formation of surface O vacancies and coordinatively unsaturated metal atoms. This disruption may spatially separate surface O species and metal atoms, potentially serving as FLPs. This structural insight suggests the possibility of the formation of CAUs, including FLPs and CUSs.

Energy-dispersive spectroscopy (EDS) mapping combined with aberration-corrected scanning transmission electron microscopy (AC-STEM) was utilized to investigate the spatial distribution of Ni atoms and atomic-scale defects (Fig. 2a–i). The low-magnification high-angle annular dark field (HAADF, Fig. 1a) image and EDS maps (Fig. 1b–e) show a homogeneous distribution of incorporated Ni atoms within the 1.8 Ni/BWO nanosheets. Aligning the 1.8 Ni/BWO nanosheet along a [001] zone axis (Supplementary Fig. 6) allowed the resolution of metal (Bi and W) and O atoms in the atomic scale integrated differential phase contrast (iDPC) image (Fig. 2f). The atomic columns, characterized by bright and dark contrasts, correspond to surface metal cations and O anions, respectively. The magnified view from the white box (Fig. 2g) demonstrates intensity fluctuations within the atomic columns. Line profiles (Fig. 2h, i) extracted from regions 1 and 2 in Fig. 2g confirm that the incorporated Ni atoms replace W/Bi atoms, resulting in a darker contrast and the formation of multiple oxygen vacancies, consistent with earlier observations. These atomic-scale microstructural changes serve as evidence for the creation of $O_Vs$ and FLPs by Ni doping in 1.8 Ni/BWO nanosheets. Some $O_Vs$ are present in the area without Ni dopants. These may be attributed to the spontaneous formation of $O_Vs$ on $Bi_2WO_6$ monolayer nanosheets.

The surface chemical states of each element were determined by X-ray photoelectron spectroscopy (XPS) spectra. W $4f$ XPS spectra (Fig. 3a) have two main peaks at 35.8 and 38.0 eV, which are attributed to the $4f_{7/2}$ and $4f_{5/2}$ of $W^{6+}$. Compared with BWO, two peaks at 35.3 and 37.5 eV assigned to $W^{5+}$ are fitted after doping with Ni[4,33], indicating that part of coordinatively saturated $W^{6+}$ atoms are mediated to

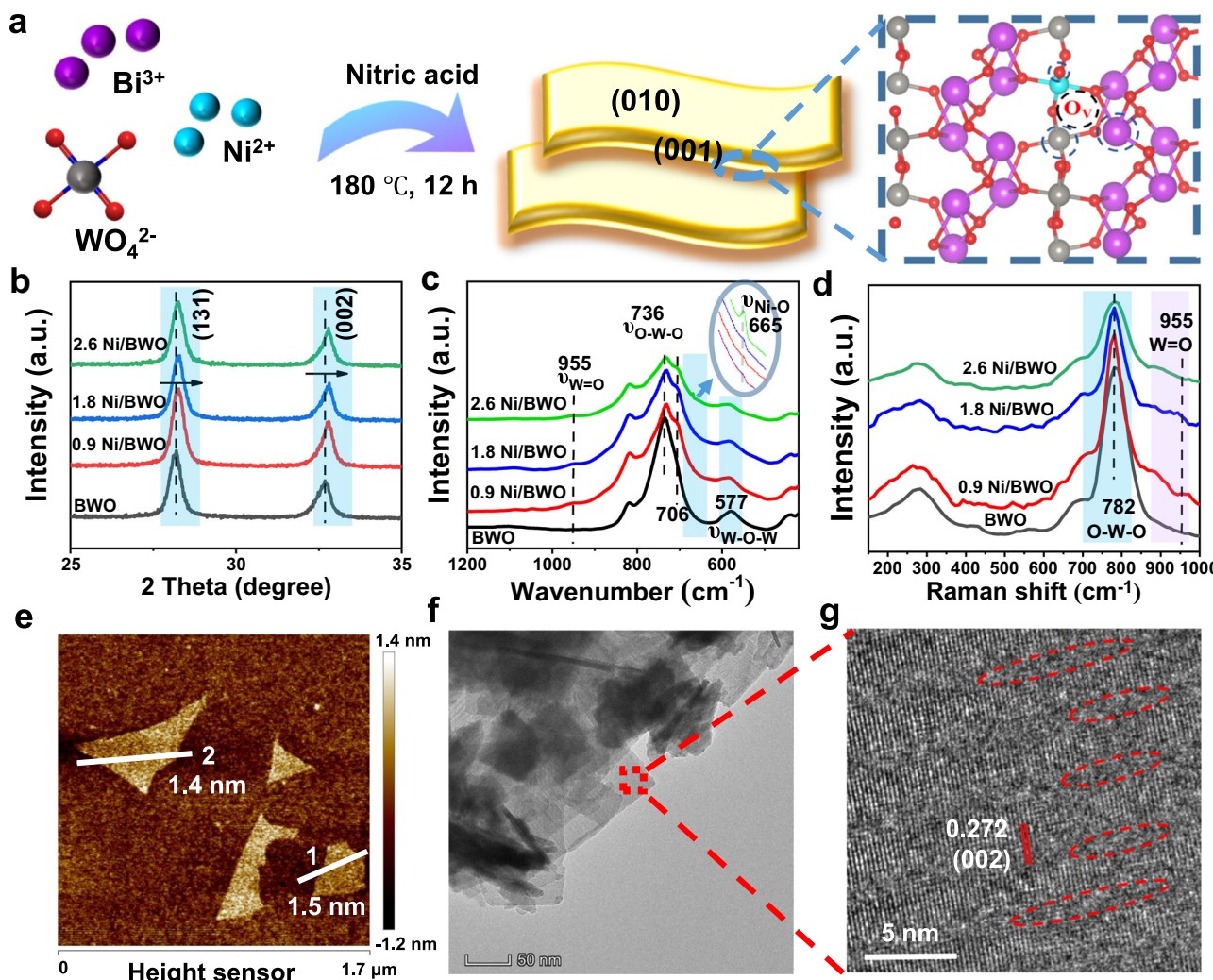

**Fig. 1 | Synthesis diagram and crystal phase structure and morphology analysis. a** Schematic diagram for the synthesis of Ni/BWO, XRD patterns, 2θ from 25° to 35° (**b**), FT-IR spectra (**c**), and Raman spectra (**d**) of the samples. AFM image (**e**), TEM (**f**), and HRTEM (**g**) image of 1.8 Ni/BWO.

unsaturated W[5+]. Detailed fitting results (Supplementary Table 2) show that excessive introduction of Ni only reduces the proportion of W atoms and does not induce more unsaturated W. In addition, Fig. 3b exhibits similar characteristic peaks of Bi[3+] for all the samples[34]. Three O species are fitted in the O 1s XPS spectra (Supplementary Fig. 7a) with binding energies of 529.5, 531.0, and 532.1 eV, corresponding to the lattice oxygen ($O_L$), $O_V$ and hydroxyl groups ($O_{OH}$), respectively[35,36]. The $O_L$ ratio (Supplementary Table 3) decreases after Ni doping, confirming that part of the lattice oxygen atoms are lost. The formation of $O_V$ is also confirmed by the EPR spectra (Fig. 3c)[27,35,37]. Interestingly, the characteristic Ni 2p XPS signal (Supplementary Fig. 7b) cannot be detected[38], but its signal is recorded utilizing cluster etching to remove the surface atoms of 1.8 Ni/BWO (Supplementary Fig. 7c). Thus, Ni is doped into the crystal lattice of BWO rather than into the surface. When the Ni is loaded onto the surface of BWO by photo-deposition, the Ni XPS signal is recorded, which also confirms the above view (Supplementary Fig. 8).

The fine structure change was analyzed by X-ray absorption fine structure (XAFS) spectroscopy. BWO and 1.8 Ni/BWO both exhibit similar absorption curves in the W $L_3$-edge XANES spectra (Supplementary Fig. 9a). The main peak intensity for 1.8 Ni/BWO is reduced in the extended X-ray absorption fine structure (EXAFS) spectra (Fig. 3d) showing that the ordered lattice arrangement of W is broken after doping with Ni[4,39]. The peaks in the correlation distance of 1–2 Å are

attributed to the signal of W-O, and the peaks in the range of 2.5–3 Å are correspond to the distance between two W atoms[39,40]. After doping with Ni ions, the peaks in the range of 2.5–3 Å are split into two peaks, manifesting the replacement of W by Ni. Meanwhile, the coordination number of W atoms decreases for the 1.8 Ni/BWO (Supplementary Fig. 9b and Supplementary Table 4). The substitution of W by Ni atoms induces the loss of some lattice O atoms, which would produce unsaturated W atoms and make the spatial separation of surface metal atoms and O species around Ni atoms, facilitating the formation of FLPs. Since the unsaturated W atoms and FLPs are both mediated by Ni dopants, they should be easily cascaded into an active unit, thus forming CAUs.

To better illustrate the construction of CAUs, the structure models of Ni/BWO were simulated and optimized by DFT calculation. According to the experimental results, we established the structure models of BWO (001) and Ni/BWO (001), while the BWO (010) and Ni/BWO (010) were also simulated for comparison. As shown in Supplementary Fig. 10a–h, the optimized structure model of BWO (010) and Ni/BWO (010) consists of $[BiO]^+$–$[WO_4]^{2-}$–$[BiO]^+$ links in the longitudinal direction from the side view. The top layer consists of Bi atoms, while Ni atoms are doped into the secondary layer. For the BWO (001) and Ni/BWO (001) models, the links of $[BiO]^+$–$[WO_4]^{2-}$–$[BiO]^+$ are in a landscape orientation (Supplementary Fig. 11a–h). The top layer consists of O atoms and the secondary layer of metal atoms. In

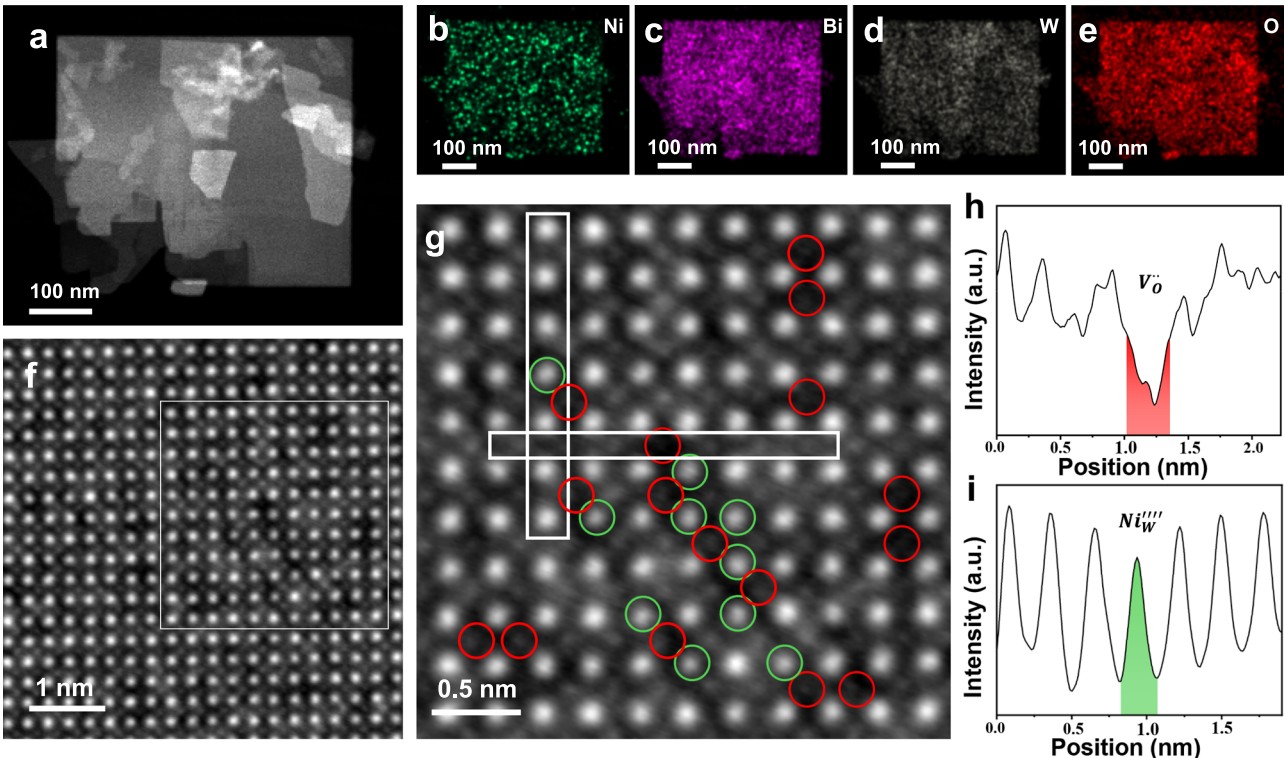

**Fig. 2 | Atomic scale composition and structure analysis.** HAADF image (**a**) and the corresponding EDS maps (**b**–**e**) of 1.8 Ni/BWO sample. The atomic-resolution iDPC image is displayed in (**f**), where the enlarged region from the white box shows the vacancy of O ion (highlighted by red circles) and the substitution of W by Ni (**g**). The line profiles of O (**h**) and W (**i**) signals are extracted from the white boxes in (**g**).

addition, the charge density differences shown in Fig. 3e show that Ni has a higher electron density while the electron density of Bi atoms around Ni is reduced. This is attributed to the surface charge rearrangement as the substitution of W atoms by Ni atoms. The $O_V$ formation energies on the (010) and (001) faces of pristine BWO and Ni/BWO are shown in Fig. 3f. The $O_V$ formation energies on the (010) and (001) faces of pristine BWO are 0.87 and 0.39 eV, respectively, while they change to −1.00 and −1.11 eV after Ni doping. These results confirm that the formation of $O_V$ is easier on BWO (001) and that Ni dopants apparently facilitate the formation of $O_V$, which is consistent with the results of experiments.

It's worth noting that the surface microstructures of pristine BWO and Ni/BWO have some differences. As shown in Fig. 3g, h and Supplementary Fig. 10c, d, the Ni atoms are doped in the secondary metal layer of the (010) surface, so the induced $O_V$ is difficult to make the spatial separation of Bi and O atoms on the top layer. Therefore, FLPs sites are hardly formed on the Ni/BWO (010). Since the W atoms are in the interior, it is difficult to form the CAUs on the (010) facet of Ni/BWO. Interestingly, CAUs containing FLPs and unsaturated W atoms can be formed on the Ni/BWO (001) surface. As shown in Fig. 3i, j and Supplementary Fig.11c, d, when the W atoms are replaced by Ni atoms, the lattice distortion, and $O_V$ are formed, which causes the nearby Bi and O atoms to separate in space, forming FLPs. These acid and base sites in the FLPs are not directly linked and thus provide catalytic active centers for activation of the C−H bonds. Meanwhile, unsaturated W atoms are also formed through the mediation of Ni dopant. Moreover, although $O_V$s are formed on pristine BWO (010) and (001) faces (Supplementary Fig. 10 e, f and Supplementary Fig. 11 e, f), the spatial separation of Bi and O atoms is difficult without Ni dopant. The formation of CAUs requires Ni dopants as switches. The Bi atoms around Ni dopants have different electron densities (Supplementary Fig. 12), which may be more active than those on pristine BWO[16].

## Photocatalytic oxidation of toluene (TL)

The series of samples were used for the photocatalytic oxidation of TL. Figure 4a shows that the performance for the photocatalytic TL oxidation is significantly improved after the introduction of Ni. Especially, 1.8 Ni/BWO exhibits the highest conversion rate of TL (4560 μmol g$^{-1}$ h$^{-1}$) which is 4.5 times higher than BWO (1020 μmol g$^{-1}$ h$^{-1}$). The nanosheet thickness, specific surface area, and energy band structure (Supplementary Fig. 13, 14) of 1.8 Ni/BWO and BWO are similar, thus the different performance is attributed to the construction of CAUs. To better compare the important role of CAUs, BWO-bulk, BWO-Ov, Ni/BWO-surface, and BiOCl were prepared. The characterization details are shown in Supplementary Figs. 16–18. BWO-bulk only shows the TL conversion rate of 200 μmol g$^{-1}$ h$^{-1}$, highlighting the structural advantage of the monolayer nanosheet. When $O_V$ is induced on BWO by heat treatment, the TL conversion rate of BWO-$O_V$ is improved to 1320 μmol g$^{-1}$ h$^{-1}$, confirming that $O_V$ promotes the photocatalytic oxidation of TL. However, the $O_V$ induced by heat treatment is random and the formation of CAUs is difficult without Ni dopant according to the DFT results, causing a low improvement of catalytic performance. When the Ni is loaded on the surface of BWO, there are almost no $O_V$ and unsaturated W as well as FLPs. Accordingly, the Ni/BWO-surface shows a TL conversion rate of 1020 μmol g$^{-1}$ h$^{-1}$, confirming the close correlation between the TL oxidation performance and the CAUs. In addition, BiOCl with $O_V$ exhibits only 90 μmol g$^{-1}$ h$^{-1}$ conversion rate of TL, also indicating that only Bi atoms and $O_V$ cannot realize the effective TL oxidation. There is a synergistic effect between surface FLPs and unsaturated W atoms, which significantly promotes the photocatalytic oxidation of TL.

The apparent quantum efficiencies (AQE) of these samples are shown in Supplementary Table 5. Typical 1.8Ni/BWO shows the highest AQE of 6.01% with a turnover frequency (TOF) about 2.2 h$^{-1}$. Figure 4b shows that the main product for the TL oxidation over 1.8Ni/BWO is benzaldehyde (BD). A small number of benzyl alcohol (BA) could be

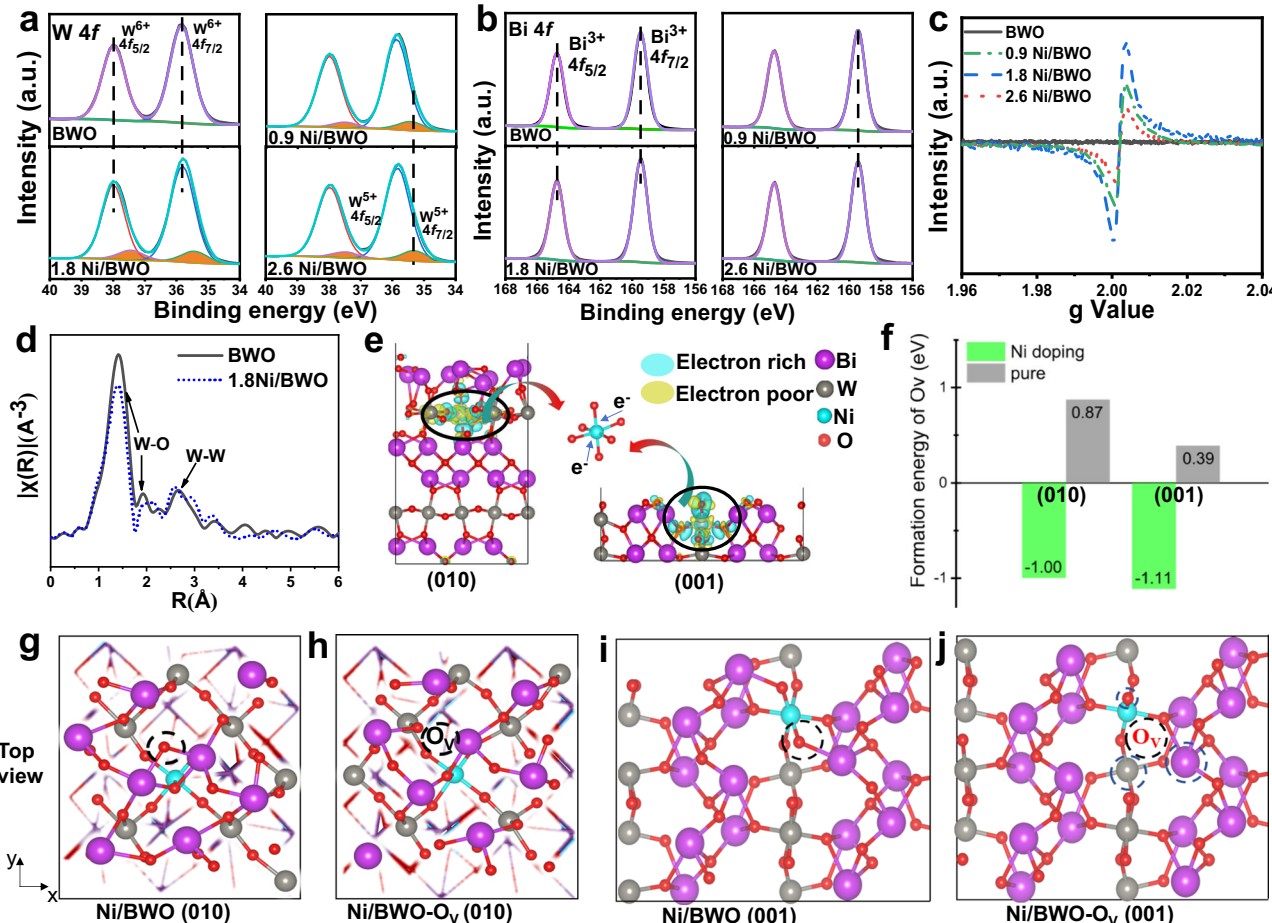

**Fig. 3 | The characterization and DFT calculations of CAUs.** XPS spectra of W4f (**a**) and Bi 4f (**b**), EPR spectra (**c**) of the prepared samples. W $L_3$-edge XAFS spectra: $k^3$-weighted Fourier transform EXAFS spectra (**d**). The optimized structure model of Ni/BWO (010) and (001) with the charge density difference (**e**). The formation energy ($E_f$) of $O_V$ (**f**). Top view of the optimized configurations of Ni/BWO (010): (**g**) initial state and (**h**) final state of forming $O_V$; Ni/BWO (001): (**i**) initial state and (**j**) final state of forming $O_V$ and FLPs.

detected at the primary stage. As the reaction time increases, these few BA molecules are further converted to BD. The selectivity of benzaldehyde has consistently been maintained at over 90%, even under the conditions of high toluene conversion (Supplementary Fig. 19a and b). Upon complete conversion of TL, BD is further oxidized to BAC. Interestingly, the further oxidation of BD to BAC is more pronounced on BWO (Fig. 4c), possibly due to the presence of CAUs promoting the conversion of the oxidation product to BD and suppressing the further oxidation of BD to BAC in the presence of TL. From the results of gas chromatography (Supplementary Fig. 19c and d), no other liquid and gas phase products are detected. No benzaldehyde is produced in the dark, regardless of temperature (Supplementary Fig. 20a). It confirms that the oxidation of toluene is driven by light. The results of five cycle experiments (Supplementary Fig. 20b), XRD pattern (Supplementary Fig. 21a), TEM, and element mapping images (Supplementary Fig. 21b, c, and d) suggest a high stability of 1.8Ni/BWO. This photocatalyst also shows good conversion for the oxidation of other toluene derivatives (Supplementary Table 6). The construction of Ni mediated CAUs is a highly efficiency strategy to improve the photocatalytic performance of TL oxidation compared to the reported studies (Supplementary Table 7).

## Study on structure-activity relationship
To understand the better performance of the 1.8 Ni/BWO, further experiments and DFT calculations were investigated. In situ Diffuse Reflectance Infrared Fourier Transform (DRIFT) spectroscopy was

applied to trace the adsorption process of $O_2$ and TL molecules. Figure 5a shows two absorption peaks appeared at 1476 cm$^{-1}$ and 1290 cm$^{-1}$, corresponding to the vibrational peaks of adsorbed oxygen and the vibration model of superoxide O−O in surface-coordinated oxygen complexes, respectively[41,42]. In comparison, BWO exhibits a weaker signal (Fig. 5b), suggesting that the CAUs of 1.8 Ni/BWO enhance the adsorption of $O_2$. As the calculated adsorption energies of $O_2$ on unsaturated Bi, Ni, and W sites are 0.31, 0.14, and −0.10 eV, (Supplementary Table 8) respectively, it tends to be adsorbed on unsaturated W atoms rather than polar FLPs. Thus, unsaturated W atoms are considered to be the active centers for the activation of $O_2$ molecules, as discussed definitely in the recent study[35]. $O_2$-TPD spectra (Fig. 5c) also shows that 1.8 Ni/BWO has a desorption peak at around 370 °C, which is related to the chemical adsorbed oxygen species, corresponding to the $O_2^-$ and $O_2^{2-}$ adsorbed on unsaturated W atoms[43]. Ni-induced the coordination-unsaturated W atoms can facilitate chemisorption of $O_2$, resulting in the formation of activated oxygen species on the surface. This is expected to play an important role in the subsequent oxygen transfer process.

Figure 5d further provides evidence for the activation of C−H. We hypothesize that the signal of TL on KBr is similar to the signal of free TL molecules. The signal of TL on KBr is still very weak after adsorption for 25 min. However, the signal peak intensity gradually increased with time on 1.8 Ni/BWO, illustrating the accumulation of the adsorbed TL. The peaks at 3084, 3064, and 3028 cm$^{-1}$ are attributed to the stretching vibration of the C−H bond of the benzene ring, while the

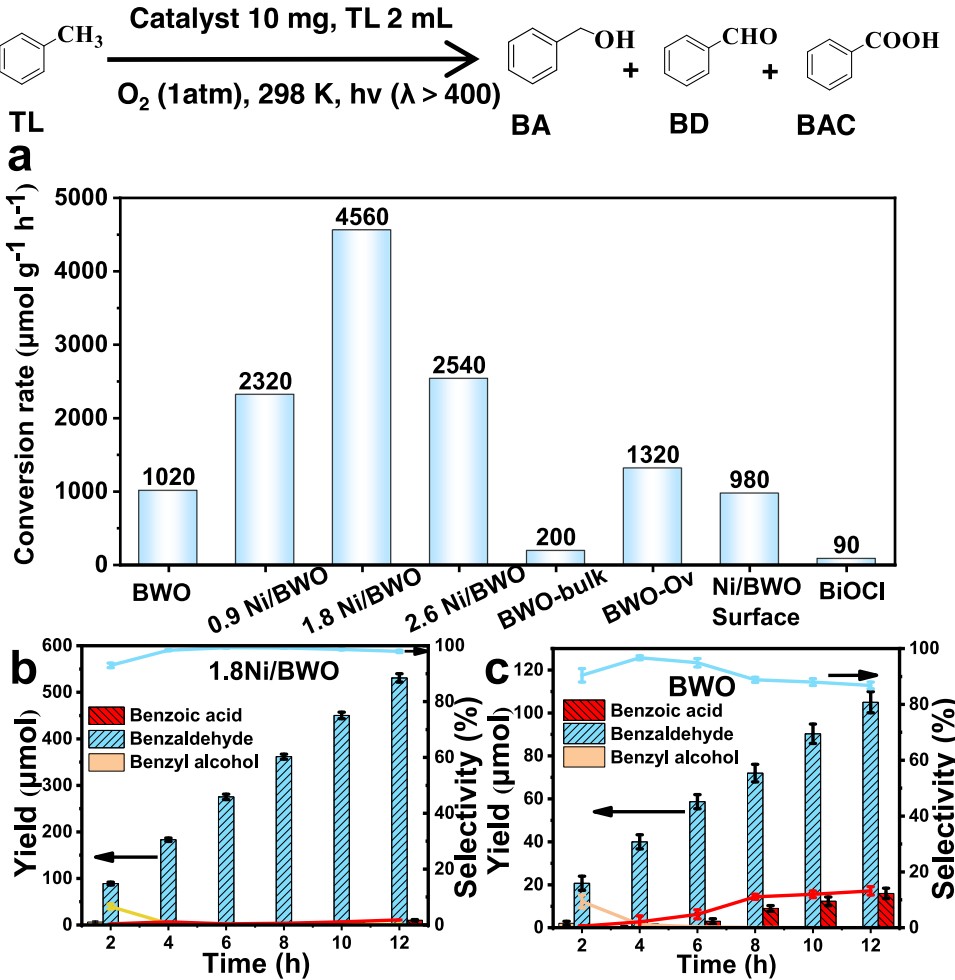

**Fig. 4 | Photocatalytic performance for toluene oxidation.** The photocatalytic oxidation of TL over different photocatalyst (**a**), The time-dependent process of the oxidation of TL over 1.8Ni/BWO (**b**) and BWO (**c**), error bars are mean ± SD based on three repeat experiments.

peaks at 2958, 2927, and 2854 cm⁻¹ are the stretching vibration of −CH₃[3,44]. Obviously, the characteristic stretching vibration peaks of the C−H bond in the benzene ring have an unnoticeable shift while the peaks of the −CH₃ stretching vibration gradually shift to 2944, 2920, and 2875 cm⁻¹, respectively, demonstrating that −CH₃ is chemisorbed and activated. In comparison, the activation of the C−H bonds on BWO is weak (Fig. 5g). This result implies that the generated surface FLPs are active centers for chemisorbing C−H bonds. In addition, the force constant (k) quantization (Supplementary Table 9) shows two C−H bonds are obviously weakened while one C−H bond is strengthened[45], indicating that the products on FLPs may be easily converted to BD.

The photocatalytic oxidation process of TL and the possible intermediates were traced by in situ DRIFTS under visible light. As shown in Fig. 5e, the characteristic peaks of the −OH stretching vibration (1230 cm⁻¹), −OH in-plane bending vibration (1314, 1340 cm⁻¹), and −C=O stretching vibration (1646, 1680 cm⁻¹) are recorded, indicating the formation of BA and BD[44,46,47]. Considering that the -C=O bond of the carboxyl group also has a characteristic peak at 1680 cm⁻¹ [48], we speculate that some BAC may be formed gradually over time. The −C=O stretching vibration peaks grow rapidly, indicating that BD is the main product during the photocatalytic process. The further oxidation of BAC to CO₂ cannot proceed because no signal of CO₂ was detected at the wavenumber 2200-2400 cm⁻¹ (Supplementary Fig. 22)[47]. Moreover, the vibrational signal of chemical adsorbed oxygen complexes (1290 cm⁻¹) gradually becomes weaker, demonstrating that there is a oxygen transfer process. This process is difficult to observe using BWO as a photocatalyst

due to the lack of CAUs (Fig. 5g). The oxygen transfer process is further confirmed by the ¹⁸O₂ isotope-labeling experiment. As shown in Fig. 5f, when using ¹⁸O₂ as oxygen source, 80% of the BD molecules are labeled by ¹⁸O, suggesting that most of the benzaldehyde's O atoms come from O₂. Considering the difficulty for lattice oxygen on tungsten (W) to participate in reactions (Supplementary Fig. 23), the remaining O atoms are considered to be the chemical adsorbed oxygen species on unsaturated W atoms when the sample is exposed to air, consistent with the results of in situ DRIFTS and O₂-TPD. This result confirms that CUSs would adsorb O₂ molecules to form chemical adsorbed oxygen species and establish a special process for transferring oxygen on the surface.

In addition, the active free radicals during the reaction are investigated by Electron Paramagnetic Resonance (EPR) technology. Supplementary Fig. 24 shows all prepared samples exhibit the carbon-centered (·C₇H₇) and •O₂⁻ radical signals due to the deprotonation of TL by photogenerated holes and the reduction of O₂ by photogenerated electrons[4,49,50], No signals are detected in the dark, indicating the necessary condition of light. 1.8 Ni/BWO shows the strongest EPR signal which is consistent with its best performance in TL oxidation. The reason is that the number of defects and active sites induced varies with different mass fractions of Ni. When the mass fraction of Ni is 1.8%, it produces the highest concentration for W⁵⁺ and FLPs (Fig. 3), which is favorable for the adsorption and activation of reactants. Surface defects also facilitate the separation of photogenerated electron-hole pairs (Supplementary Fig. 25)[51], thereby enhancing photocatalytic performance. The free radical quenching experiments

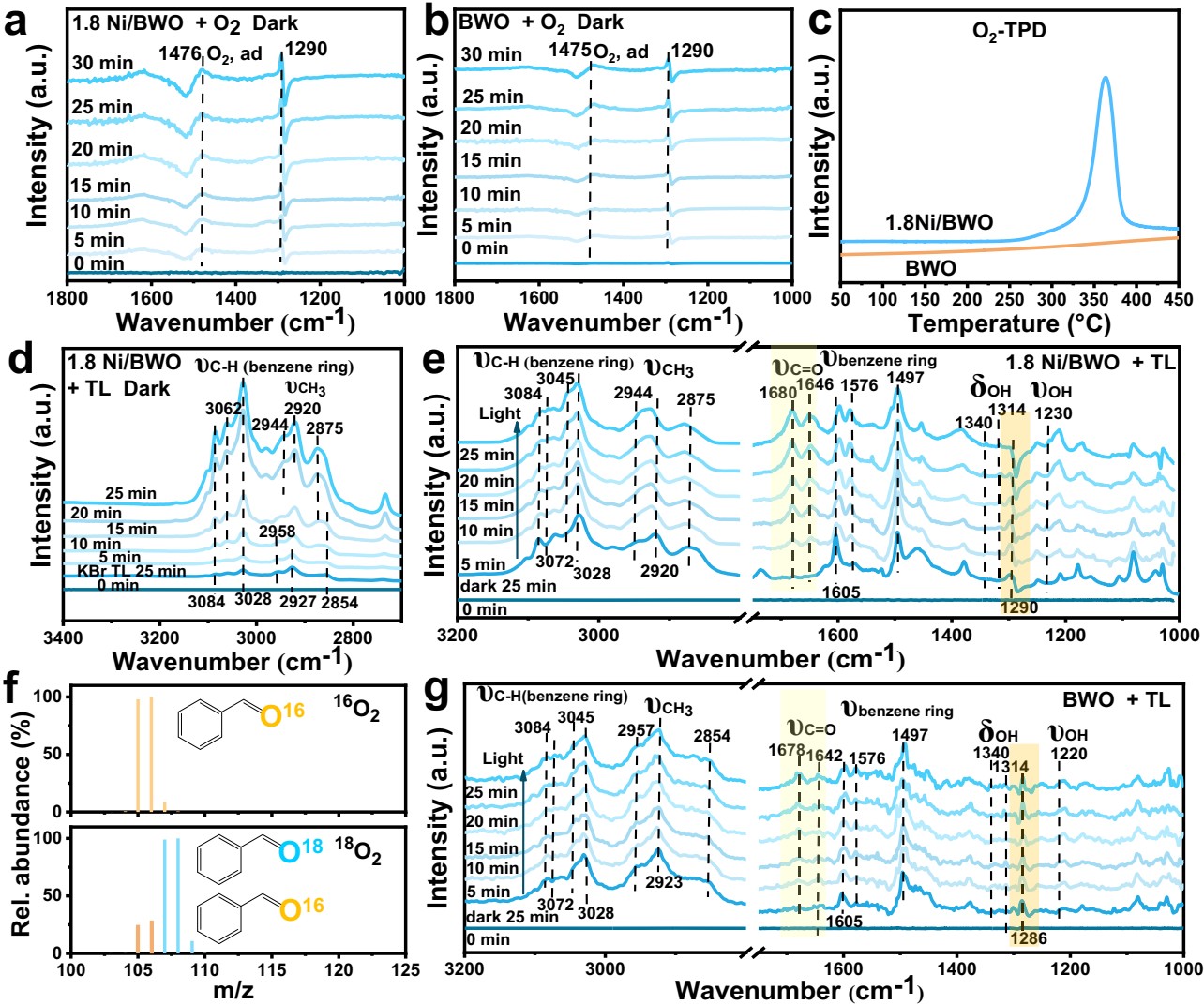

**Fig. 5 | The adsorption behavior of toluene and O₂, and the reaction process analysis.** Time-dependent in situ DRIFT spectra of 1.8 Ni/BWO (**a**) and BWO (**b**) in the dark under O₂ atmosphere. O₂-TPD spectra of the prepared samples (**c**). Time-dependent in situ DRIFT spectra of 1.8 Ni/BWO in the dark under toluene and O₂ atmosphere (**d**). Time-dependent in situ DRIFT spectra of 1.8 Ni/BWO (**e**) and BWO (**g**) in a toluene and O₂ atmosphere under visible light irradiation. Mass spectra of BD produced in $^{16}O_2$ and $^{18}O_2$ atmosphere over 1.8 Ni/BWO (**f**).

reveal that the deprotonation of TL by photogenerated holes is the key step for the oxidation of TL and $\cdot O_2^-$ radicals are considered as one of the active O species in the subsequent oxidation process and another O transfer pathway also exists (Supplementary Fig. 24c).

The DFT simulations are also performed to reveal the surface O transfer process based on the optimized Ni/BWO (001) model. As shown in Fig. 6a, structure I is the model of Ni/BWO (001). When the O₂ molecule is added, this model tends to adsorb the O₂ molecule with a binding energy of −0.1 eV (model II). The O₂ molecule is then activated via two W···O coordination. On the one hand, some of these activated O₂ molecules can be directly reduced to $\cdot O_2^-$ radical species by the photo-excited electrons (Supplementary Fig. 24b). On the other hand, as the distance between two O atoms is stretched by W atoms in Fig. 6a TS1, the chemical adsorbed oxygen species are eventually broken (model III). Interestingly, the O atoms from the broken chemical adsorbed oxygen species in model III can easily deliver to Ni atoms (TS2). The energy barrier for this surface O delivery process from model III to model IV is sufficiently low (0.59 eV) and easy to overcome. The transferred active O atoms can be used for the formation of BD and H₂O. This surface active O transfer channel is more efficient because it occurs in a CAU.

Meanwhile, the calculated activation path of the C−H bond in TL on Ni/BWO (001) is compared with that on Ni/BWO (010), BWO-O$_V$ (010) and BWO-O$_V$ (001) in Fig. 6b. "2* + O$_V$" represents those two active atoms around O$_V$ selected to combine with the C and H atoms of TL. The subsequent $^*+^*C_7H_8$ (TL) dehydrogenation to $^*C_7H_7$-H$^*$ undergoes a transition state (TS). The intermediate $^*C_7H_7$ has been confirmed by EPR experiments (Supplementary Fig. 24a). The corresponding geometries and energies are shown in Fig. 6c. The energy barriers to be overcome for the dehydrogenation of toluene on Ni/BWO (001) (TS₄), Ni/BWO (010) (TS₃), BWO-OV (001) (TS₂), and BWO-OV (010) (TS₁) are 0.23, 0.28, 1.02, and 1.16 eV, respectively. The energy barriers on BWO-OV (001) and BWO-OV (010) are significantly lower than those on Ni/BWO (001) and Ni/BWO (010), indicating that the Bi and O atoms on Ni/BWO (001) and Ni/BWO (010) are more active due to the regulation of the Ni dopants. Particularly, the TL molecules adsorbed on Ni/BWO (001) are easily dehydrogenated due to the formation of Bi···C and O···H coordination between the FLPs and the C−H bond. The transient charge density difference image of TS4 (Fig. 6d) shows that the Bi···C and O···H coordination cause the electrons of H in C−H bonds to transfer to O atoms and the electrons of Bi to shift to C, resulting in the effective activation of C−H bonds.

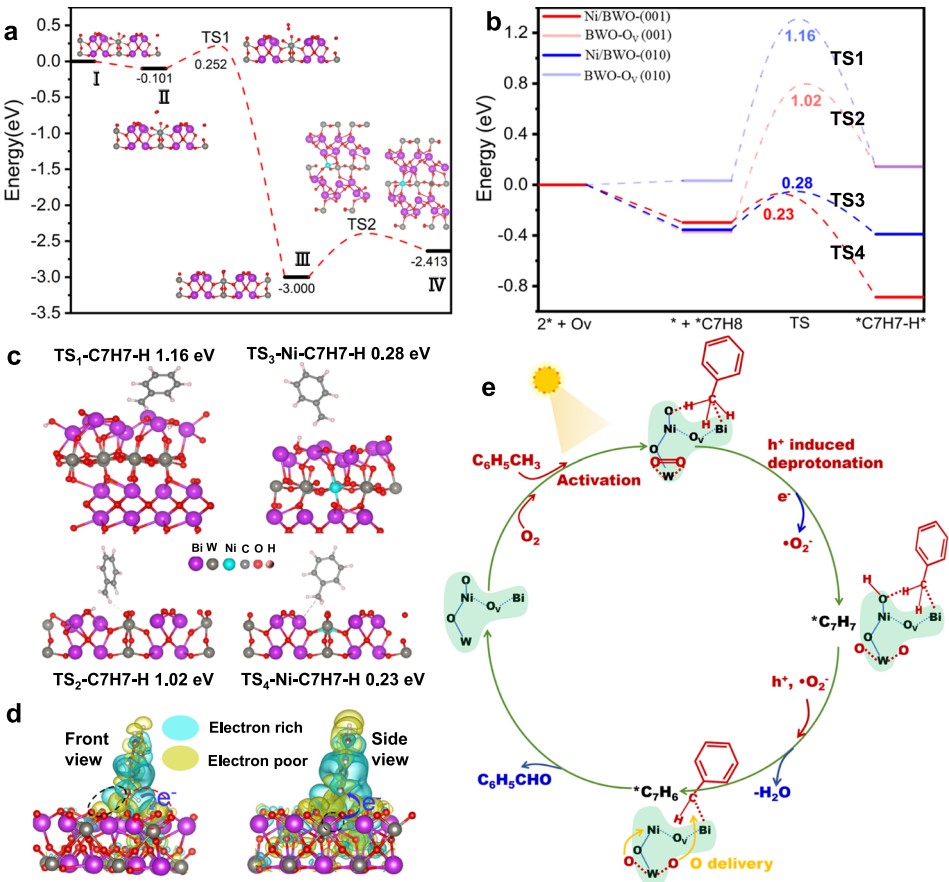

**Fig. 6 | DFT calculations of the C–H activation and oxygen transfer pathway, and possible mechanism.** The activation and delivery of $O_2$ with the calculated energy profiles on the optimized structural models of Ni/BWO-(001) (**a**). Calculated energy profiles of toluene adsorbed on different structure models (**b**), corresponding geometries and energies of $TS_1$, $TS_2$, $TS_3$, and $TS_4$ (**c**), Transient charge density difference image of toluene adsorbed on the FLPs of Ni/BWO-(001) (**d**), The possible mechanism for photocatalytic oxidation of toluene on 1.8 Ni/BWO (**e**).

Furthermore, since doping can lead to a redistribution of charge around neighboring atoms[52], we considered the influence of potential surrounding sites (such as Ni and Bi site) and found that the current FLP adsorption system for the C–H has the lowest energy (Supplementary Fig. 26a). The Bi···C and O···H coordination as bridges for the transfer of photogenerated holes from the Bi and O atoms to C–H bonds, which significantly promotes the deprotonation of the C–H bonds. After the dehydrogenation of TL, the O atoms in the FLPs could combine with H atoms to form surface –OH (Supplementary Fig. 26b). The corresponding valence states of the spatially separated Bi and O atoms show little change (Supplementary Table 10) and can continue to be used as FLPs to dehydrogenate the $^{*}C_7H_7$ intermediate. More surface hydroxyl groups on 1.8 Ni/BWO (XPS results) also indicate the highest levels of FLPs on 1.8 Ni/BWO, which promotes the activation and dehydrogenation of the C–H bond. The surface –OH combines with the H atom in the C–H bond to form a desorbed $H_2O$ molecule, leaving a lowly coordinated Ni atom and the $^{*}C_7H_6$ intermediate. The delivered O atoms in model IV of Fig. 6a would further transfer to the $^{*}C_7H_6$ intermediate to form benzaldehyde. The desorption energies of benzaldehyde on Ni/BWO-(001) and BWO-(001) are 0.41 and 0.69 eV (Supplementary Fig. 26d), respectively, indicating the easier desorption of benzaldehyde on Ni/BWO-(001). This provides a reasonable explanation for the better selectivity of benzaldehyde on 1.8 Ni/BWO than that on BWO.

Based on the experimental results and the DFT simulations, we find a special surface pathway for the photocatalytic toluene oxidation induced by the Ni dopant mediated CAUs (Fig. 6e), which is different from the traditional photocatalytic mechanism (Supplementary Fig. 27). Firstly, the $O_2$ and TL molecules would be activated on the unsaturated W atoms and FLPs, respectively. Then, some of the activated $O_2$ and TL molecules can be converted to $\cdot O_2^-$ and $^{*}C_7H_7$ radicals under the attack of photogenerated electrons and holes. Moreover, some activated oxygen species can be directly broken on unsaturated W atoms, while a C–H bond in the $-CH_3$ group is broken with the formation of $^{*}C_7H_7$ intermediates and a surface hydroxyl group. The $^{*}C_7H_7$ intermediates can be further oxidized to $^{*}C_7H_6$ and release an $H_2O$ molecule by photogenerated $h^+$ and $\cdot O_2^-$ radicals. Finally, active O species are efficiently transferred in CAUs to oxygenate the $^{*}C_7H_6$ intermediates to BD and replenish the O consumption of dehydration for starting the next cycle.

In summary, we have successfully developed monolayer Ni-doped $B_2WO_6$ nanosheets (Ni/BWO) as a highly efficient photocatalyst for the selective oxidation of toluene to benzaldehyde. The experimental characterization clarified that Ni dopants mediated cascaded active units (CAUs) at the atomic scale, including unsaturated metal W atoms and spatially separated Bi and O atoms as FLP sites. The FLPs sites are the active centers for adsorbing the C–H bonds of toluene via Bi···C and O···H coordination, while $O_2$ is adsorbed on the unsaturated metal W sites. It builds bridges for the effective transfer of photogenerated carriers, promoting the subsequent process of toluene oxidation by light-induced active species. In addition, the surface FLPs and unsaturated W sites also establish a special route for the oxidation of C–H bonds, including the dehydrogenation of C–H bonds, the activation of $O_2$, and the efficient delivery of surface O. Therefore, the representative 1.8 Ni/BWO exhibits a significantly enhanced toluene conversion rate of 4560 μmol g$^{-1}$ h$^{-1}$. This work presents an experimental and

theoretical basis for designing CAUs with a synergistic effect via Ni doping.

## Methods

### Preparation of monolayer Ni doped $Bi_2WO_6$ nanosheets (Ni/BWO)

Ni/BWO with different Ni content were prepared via a one-step hydrothermal method[22]. Firstly, 1 mL of nitric acid was added to 35 ml of deionized water. Then, 2 mmol $Bi(NO_3)_3 \cdot 5 H_2O$, x (x = 0.05, 0.10 and 0.15) mmol $Ni(NO_3)_2 \cdot 6H_2O$ and (1-x) mmol $Na_2WO_4 \cdot 2H_2O$ were dissolved in the above nitric acid solution under stirring. After mixing uniformly, the mixture was transferred into a 50 mL Teflon-lined stainless-steel autoclave, and heated at 180 °C for 12 h. After cooling to room temperature, the samples were washed with deionized water. Then, the sample were dried at 100 °C for 24 h. The actual mass fraction of Ni was ascertained by Inductively Coupled Plasma Mass Spectrometry (ICP-MS). As shown in Supplementary Table 1, the actual mass fraction of Ni in the series samples was 0.9%, 1.8%, and 2.6%. The corresponding samples were named as 0.9 Ni/BWO, 1.8 Ni/BWO, and 2.6 Ni/BWO. The monolayer $Bi_2WO_6$ nanosheets (BWO) were prepared via the same route without adding $Ni(NO_3)_2 \cdot 6H_2O$. The bulk $Bi_2WO_6$ (BWO-bulk) were also prepared via the same route without adding nitric acid and $Ni(NO_3)_2 \cdot 6H_2O$.

### Characterization

The crystal phases of the powder samples were analyzed on a Bruker D8 Advance X-ray diffractometer with CuKα radiation (λ = 0.15406 nm) at 40 kV and 40 mA in the range of 5−60°. Inductively coupled plasma optical emission spectrometry (ICP-OES) was recorded on a PerkinElmer (Avio 200). The field emission scanning electron microscope (SEM) images of the powder samples were obtained on a FEI Quanta 200 F electron microscope at an acceleration voltage of 10 kV. Higher-resolution transmission electron microscopy (HRTEM), transmission electron microscopy (TEM), and element mapping images were recorded on a JEOL model JEM2010 EX microscope at an accelerating voltage of 200 kV. The atomic force microscopy (AFM) images of the samples were collected using a Bruker Dimension Icon with a scanning frequency of 512. The Raman spectra of the samples were obtained using a Renishaw Invia Raman microscope. X-ray photoelectron spectroscopy (XPS) tests of the samples were carried out on a Thermo ESCALAB 250Xi photoelectron spectroscopy with monochromatic Al Ka Radiation (E = 1486.2 eV) at $3.0 \times 10^{-10}$ mbar. An internal and external neutralizing gun was selected to mitigated surface charging. All binding energies were calibrated using the C1s peak at 284.6 eV. The Brunauer−Emmett−Teller (BET) specific surface areas of the series of samples were investigated using a Micrometrics ASAP 2020 system on an Autosorb-1C-TCD physical adsorption instrument (American Quantach-Rome). The UV−vis diffuse reflectance spectra (UV−vis DRS) were determined on a Cary 500 Scan UV−vis spectrophotometer (Varian) using $BaSO_4$ as a reflectance standard. The scan wavelength was 200−800 nm with a scan speed 600 nm/min. The Mott-Schottky test, electrochemical impedance spectroscopy (EIS), and photocurrent (PC) measurements were carried out using a Zahner electrochemical workstation. Steady-state photoluminescence spectra (PL) and time-resolved photoluminescence spectra (time-resolved PL) were detected on a HORIBA Fluorolog-3 Fluorescence Spectrometer. $O_2$ temperature programmed desorption ($O_2$-TPD) spectra were test on a AutoChem II (Micromeritics Instrument Corp.)

### Photocatalytic oxidation of toluene

The photocatalytic oxidation of TL was conducted in a 10 mL quartz reaction tube. (Supplementary Fig. 15) 10 mg of the prepared samples, 2 mL TL were added to the reaction tube under stir. $O_2$ was continuously introduced into the reaction tube to remove air. Then the reaction tube was connected with a $O_2$ balloon. After 30 min stir, the mixture was irradiated by a 300 W Xenon lamp (Beijing Perfectlight Technology Co. Ltd., PLS-SXE300D) with a 400 nm cut-off filter. After the reaction, the gas phase was determined by a gas chromatography (GC-7890B, Agilent) with a TCD, an FID detector, and two connected columns (MolSieve 5 A and HP-PLOT Q). The solid sample was removed and the solution was analyzed using a Shimadzu Gas Chromatograph (GC2014C) with a Shimadzu SH-RTX-S column (film thickness 0.25 μm; length 30 m; inner diameter 0.32 mm). The used catalyst is cleaned with methanol for three times and collected after drying in 60 °C for 6 h to conduct the cycle experiments. The products of isotope-labeling experiment are test using a Gas chromatograph mass spectrometer (GCMS-QP2020 NX).

### Density functional theory calculation

In this work, the DFT method with spin polarization was employed for all calculations using the Vienna Ab-initio Simulation Package (VASP 6.3.0). To accurately represent the interaction between core electrons, we utilize Projector-augmented wave (PAW) pseudopotentials and employ the Perdew−Burke−Ernzerhof (PBE) exchange-correlation functional[53] of generalized gradient approximation (GGA)[54]. Additionally, the DFT-D3 method with Becke−Johnson (BJ) damping[55] was incorporated to approximate the dispersion effect. To eliminate the interaction between atoms in adjacent periodic layers, a vacuum layer of up to 15 Å was added to the BWO surface (010 and 001) on the z-directions. A standard Monkhorst−Pack grid sampling with a $2 \times 2 \times 1$ k-point mesh was used in all calculation. The plane wave basis set with a cut-off energy of 400 eV is used and the calculated results were based on the convergence of electron step (EDIFF = $10^{-4}$ eV) and ion step (EDIFFG = $-0.05$ eV/Å), which successfully achieved a balance between high computational efficiency and high precision. The combination of climbing image nudged elastic band (CI-NEB)[56] and dimer[57] was cleverly employed for the search of transition state structures. And all transition state structures found were subjected to frequency calculations to verify if there were any imaginary frequencies corresponding to the vibration of chemical bonds breaking and forming.

### Reporting summary

Further information on research design is available in the Nature Portfolio Reporting Summary linked to this article.

## Data availability

All data supporting the findings of this study are available in the article and its Supplementary Information. Source data are provided with this paper.

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

## Acknowledgements

This work was supported by the National Natural Science Foundation of China (Grant No. 22272026, L.W., 22373017, L.S., 22272024, Z.Y. and 52222102, Z.Y.), and the 111 Project (D16008). S.L. thanks the "Chuying Program" for the Top Young Talents of Fujian Province. Numerical computations were performed on the Hefei Advanced Computing Center and Supercomputing Center of Fujian. The authors acknowledge the facility resources from Electron Microscopy Center of Fuzhou University.

## Author contributions

S. Lin and L. Wu conceived and led the project. Y. Shi, Z. Wang, and Y. Song designed and carried out the experiments. S. Lin and P. Li designed the simulations and P. Li carried out the DFT simulations. L. Wu, Z. Yu, Y. Shi, Y. Tang, H. Chen contributed to data analysis. P. Li wrote the drafts for the theoretical part. Y. Shi wrote the original draft. L. Wu, S. Lin, Z. Yu, J. C. Yu, X.Z. Fu reviewed and validated the manuscript. All authors discussed results and provided comments during the manuscript preparation.

## Competing interests

The authors declare no competing interests.
