## [Peer Review File · Nature Communications]

Photocatalytic toluene oxidation with nickel-mediated cascaded active units over Ni/Bi₂WO₆ monolayersREVIEWER COMMENTS

Reviewer #1 (Remarks to the Author):

The authors designed a Ni-doped monolayer Bi₂WO₆ nanosheets, achieving photocatalytic toluene selective oxidation under mild conditions. Structural characterizations and DFT studies confirm the lattice W atom substitution by Ni dopants. Mechanism studies demonstrated that Ni dopants played a vital role in the C–H bond activation and the active O transfer that boost the toluene conversion. Considering recent research trends, this study can be of interest to many researchers. However, there are some critical issues that should be addressed for the publication of this work.

1. The authors claim that “The catalyst with a Ni mass fraction of 1.8% exhibits excellent toluene conversion rates and high selectivity towards benzaldehyde” (Abstract, line 27). Notably, no data regarding the selectivity of benzaldehyde was provided. Additionally, how is the photocatalytic performance and product distribution at high toluene conversion?
2. Figure 2 lacks error bars, and the label of Figure 2e is incorrect.
3. Oxygen transfer process should be investigated by ¹⁸O₂ isotope-labeling experiment.
4. I am concerned that the authors use in situ DRIFTS to trace the adsorption behavior of O₂ on the catalyst, given that vibrational signal of O₂ (1350~1500 cm⁻¹) and adsorbed O species (<1100 cm⁻¹) typically appears in the indiscernible fingerprint region of IR (Nat. Commun. 2023, 14:6275). A thorough re-analysis of oxygen adsorption by Raman or O₂ TPD to support the conclusion is necessary.
5. The authors also highlights that “the O atoms from the broken O₂ molecule” can be used for the formation of BD and H₂O, however, the form of O atoms involved in the reaction is unclear (surface lattice O atoms or just O₂?). Additional experiments to support the hypothesis of the reaction pathway should be included and the clarity of the mechanism can be improved.
6. Can the authors comment on how the mass fractions of Ni (0.9, 1.8, 2.6) influence the photocatalytic performance?

Reviewer #2 (Remarks to the Author):

This manuscript focuses on the photocatalytic selective oxidation of toluene via the creation of nickel-mediated cascaded active units as photocatalytic active centers. These cascaded active units include unsaturated W atoms, oxygen vacancies and Bi/O frustrated Lewis pairs (FLPs), which not only enable the efficient activation of C-H bonds and O₂, but also facilitate the transport of the active O species, thus promoting a very efficient catalytic process. Overall, the manuscript is well organized in terms of materials characterization and mechanistic studies. It is an interesting paper with novel insights for future catalyst design. The following minor issues should be addressed before acceptance.

1. The photocatalytic reaction is irradiated using a 300 W Xenon lamp. Exposure to light generates heat, how is the temperature of the catalytic system maintained? What is the effect of temperature on reaction performance?
2. The selectivity of the products and the turnover frequency (TOF) should be evaluated.

3. In the simulation of the catalytic mechanism, have the authors considered the activation of O₂ by the unsaturated Bi atom, while the O atom and the unsaturated W atom serve as the activation sites for toluene? Can the O and W atoms also serve as FLP sites for toluene activation?

4. DFT calculations indicate that by doping with Ni, the activation barrier for C-H bond activation can be as low as 0.3 eV, while the spillover of activated O atoms to the FLP site becomes the rate-determining step with a high activation energy of 0.59 eV. Have the authors considered reducing the barrier for the spillover of O atoms (e.g. by designing a type of NiM bimetallic catalyst) to achieve more efficient oxidation of toluene?

Reviewer #3 (Remarks to the Author):

Shi et al. report the nickel-mediated cascaded active units in monolayer Ni/Bi₂WO₆ for photocatalytic selective toluene oxidation. The cascaded active units (CAU) consist of unsaturated W atoms and Bi/O frustrated Lewis pairs (FLPs). A very interesting finding from the DFT and experimental results is that the CAU can promote both the activation of C-H bonds and O₂ and the transportation of active O, thereby establishing a very efficient C-H bond oxidation process. In my opinion, the creation of cascaded active units is innovative and this work was well organized. I am very impressed with this work, which provides a significant new contribution that highlights the strategy of creating CAUs with synergistic effects for photocatalytic toluene oxidation. The insights gained from this work can help accelerate the development of related photocatalysts. Therefore, I recommend the publication of this manuscript after addressing some minor issues as shown below:

1. Figure 2g shows that some OVs are present in the area without Ni dopants. What is the reason for the formation of these OVs? Do these OVs promote the photocatalytic performance.
2. It is found that 1.8 Ni/BWO has more surface hydroxyl group species than other samples from the XPS results. The influence of surface hydroxyl group species on the photocatalytic performance should be discussed.
3. What is the procedure for the regeneration of the produced catalysts during the cycle experiments?
4. The author focuses on analyzing the significance of Ni-mediated unsaturated W for the C-H activation of toluene, but seldom discusses the activation process of the phenyl group's C-H bond. The entire process of moderate oxidization of toluene to benzaldehyde also involves the activation of the C-H bond in the phenyl group. The author should supplement whether the unique active center has a similar effect on the activation of C-H bond in the phenyl group as it does on the activation of the C-H bond in toluene.

Reviewer #4 (Remarks to the Author):

Lin et al reported a series of Ni-doped monolayer Bi₂WO₆ nanosheets (Ni/BWO) with varying Ni mass fractions for the photocatalytic selective oxidation of toluene. The composition of the Ni/BWO was analyzed by ICP-MS, TEM and AFM. In the photocatalytic oxidation of toluene, a high selectivity of benzaldehyde was obtained. However, the performance of the current manuscript is not innovative

enough. So, the work needs major revisions before it can be considered for publication in Nature Communication.

1. The introduction of the second metal changes the electron configuration around the original metal. (Angew Chem Int Ed 2018, 57, 9660.) The exact role of Ni should be carefully studied. Is Ni the catalytic active site or is Ni replacing W affecting the electron configuration around W, and the catalytic active site is still W?
2. Not only comparing with the BWO-Ov, BiOCl and so on, summarizing a list including the performance of other reported catalysts for toluene oxidation is necessary.
3. Please supplement the photocatalytic oxidation properties of toluene derivatives with Ni/BWO as the photocatalyst.
4. "BA is further converted to BD and a small amount of BD is oxidized to benzoic acid (BAC)" in line 283. However, toluene can also be reported to oxidize directly to benzaldehyde (10.1021/jacs.9b09954). Please give the evidence that BA is converted to BD, not TL directly oxidized to BD.
5. Pay attention to writing norms. (Csp³-H, in line 319 and 332)

Point-by-point response to the reviewers' comments

GENERAL: We are grateful to the reviewers for all their comments, which help us greatly to improve the manuscript. We have carefully considered all of them and have revised the manuscript accordingly. The changes are highlighted in blue in the manuscript. The detailed responses to all the critical points are given below, with the original comments in italics.

REVIEWER COMMENTS

Reviewer #1 (Remarks to the Author):

The authors designed a Ni-doped monolayer Bi₂WO₆ nanosheets, achieving photocatalytic toluene selective oxidation under mild conditions. Structural characterizations and DFT studies confirm the lattice W atom substitution by Ni dopants. Mechanism studies demonstrated that Ni dopants played a vital role in the C–H bond activation and the active O transfer that boost the toluene conversion. Considering recent research trends, this study can be of interest to many researchers. However, there are some critical issues that should be addressed for the publication of this work.

1. The authors claim that “The catalyst with a Ni mass fraction of 1.8% exhibits excellent toluene conversion rates and high selectivity towards benzaldehyde” (Abstract, line 27). Notably, no data regarding the selectivity of benzaldehyde was provided. Additionally, how is the photocatalytic performance and product distribution at high toluene conversion?

Response: Thanks for your comments. Figure R1a shows the selectivity of the product at different reaction times. Reproducibility studies were conducted and error bars were added. When using 1.8Ni/BWO for photocatalytic toluene oxidation, benzyl alcohol was detected with a selectivity of 9% at the primary stage (reaction time 2h). As the reaction time increases, benzyl alcohol is further converted to benzaldehyde with a selectivity over 95%. A small amount of benzaldehyde is gradually oxidized to benzoic acid. Interestingly, the oxidation of benzaldehyde to benzoic acid is more pronounced

on BWO (Figure R1b), possibly due to the presence of CAUs promoting the conversion of the oxidation product to BD. The selectivity of benzaldehyde has consistently been maintained at over 90%. Thus, the catalyst with a Ni mass fraction of 1.8% exhibits excellent toluene conversion rates and high selectivity towards benzaldehyde (Abstract, line 27).

However, after 12 hours of reaction in toluene as the solvent, the conversion of toluene was too low to reflect the product distribution under high conversion conditions. To observe the product distribution of toluene at a high conversion, we conducted photocatalytic performance tests using acetonitrile as the solvent and 0.1 mmol of toluene as the reactant. Figure R1c and d show the results, indicating that the main product remains BD even at high TL conversion. Upon complete conversion of TL, BD is further oxidized to BAC. The results indicate that CAUs promote the conversion of the oxidation product to BD, while the further oxidation of BD to BAC is suppressed in the presence of TL.

Figure R1 The time-dependent process of the oxidation of TL over 1.8Ni/BWO (a) and BWO (b), error bars are mean \pm SD based on 3 repeat experiments. Gas phase (c) analysis for different reaction time from our online GC setup and the corresponding conversion of TL and selectivity of BD (d); reaction condition: 10 mg 1.8 Ni/BWO, 0.1 mmol TL, 1.5 mL acetonitrile, O₂.

Corresponding Reversion: The sentences “A small number of benzyl alcohol (BA) could be detected at the primary stage. As the reaction time increases, these few BA molecules are further converted to BD. The selectivity of benzaldehyde has consistently been maintained at over 90%, even under the conditions of high toluene conversion (Figure S19a and b). Upon complete conversion of TL, BD is further oxidized to BAC.” have been added to the revised manuscript.

“possibly due to the presence of CAUs promoting the conversion of the oxidation product to BD and suppressing the further oxidation of BD to BAC in the presence of TL.” has been added to the revised manuscript.

Figure R2 (a) and (b) have been added to the revised manuscript as Figure 4 (b) and (c).

Figure R2 (c) and (d) have been added to the Supplementary information as Figure S19 (a) and (b).

2. Figure 2 lacks scale bars, and the label of Figure 2e is incorrect.

Response: Thanks for your reminder. Accordingly, we have added the scale bar and revised the label of Figure 2e.

Corresponding Revision: Figure 2 has been replaced by Figure R2 shown below.

Figure R2 HAADF image (a) and the corresponding EDS maps (b-e) of 1.8 Ni/BWO sample. The

atomic-resolution iDPC image is displayed in (f), where the enlarged region from the white box shows the vacancy of O ion (highlighted by red circles) and the substitution of W by Ni (g). The line profiles of O (h) and W (i) signals are extracted from the white boxes in (g).

3. Oxygen transfer process should be investigated by $^{18}\text{O}_2$ isotope-labeling experiment.

Response: Thanks for your important suggestion. Accordingly, we conducted an $^{18}\text{O}_2$ isotope-labeling experiment to study the oxygen transfer process, as shown in Figure R3. The results show that 80% of the BD molecules are labeled by ^{18}O , suggesting that most of the benzaldehyde's O atoms come from the O_2 supplied during the experiment. The remaining O atoms are considered to be chemical adsorbed oxygen species on unsaturated W atoms when the sample is exposed to air. These oxygen species, which are adsorbed on unsaturated W atoms, will significantly contribute to establishing a new process for transferring oxygen on the surface.

Corresponding Revision: The sentences “The unique oxygen transfer process is further confirmed by the $^{18}\text{O}_2$ isotope-labeling experiment. As shown in Figure 5f, when using $^{18}\text{O}_2$ as oxygen source, 80% of the BD molecules are labeled by ^{18}O , suggesting that most of the benzaldehyde's O atoms come from O_2 . Considering the difficulty for lattice oxygen on tungsten (W) to participate in reactions (Figure S23), the remaining O atoms are considered to be the chemical adsorbed oxygen species on unsaturated W atoms when the sample is exposed to air, consistent with the results of in situ DRIFTS and O_2 -TPD. This result confirms that CUSs would adsorb O_2 molecules to form chemical adsorbed oxygen species and establish a new process for transferring oxygen on the surface” have been added to the revised manuscript.

Figure R3 has been added to the revised manuscript as Figure 5 (f).

The sentence “The products of isotope-labeling experiment are test using a Gas chromatograph mass spectrometer (GCMS-QP2020 NX).” has been added to the **Methods**.

Figure R3 Mass spectra of BD produced in $^{16}\text{O}_2$ and $^{18}\text{O}_2$ atmosphere over 1.8Ni/BWO.

4. I am concerned that the authors use *in situ* DRIFTs to trace the adsorption behavior of O_2 on the catalyst, given that vibrational signal of O_2 ($1350\sim 1500\text{ cm}^{-1}$) and adsorbed O species ($<1100\text{ cm}^{-1}$) typically appears in the indiscernible fingerprint region of IR (Nat. Commun. 2023, 14:6275). A thorough re-analysis of oxygen adsorption by Raman or O_2 TPD to support the conclusion is necessary.

Response: Thanks for your valuable comment. Our analysis of the *in situ* DRIFT spectra of oxygen adsorption requires further justification. Figure R4a shows two absorption peaks appeared at 1476 cm^{-1} and 1290 cm^{-1} , corresponding to the vibrational peaks of adsorbed oxygen and the vibration model of superoxide O-O in surface-coordinated oxygen complexes, respectively ^{r1, r2}. This suggests the activation of O_2 molecules. The intensity of these peaks increased over time, and 1.8Ni/BWO exhibited a more pronounced increasing trend than BWO. Combined with our recent studies, it is suggested that the abundant unsaturated W atoms on 1.8Ni/BWO are considered to be the active centers for the activation of O_2 molecules.

As recommended by the reviewer, we conducted O_2 -TPD test to reveal the oxygen adsorption behavior. Figure R4c shows that 1.8 Ni/BWO has a desorption peak at

approximately 370 °C, which is related to the chemical adsorbed oxygen species, corresponding to the O_2^- and O_2^{2-} adsorbed on unsaturated W atoms.^{r3} The O_2 -TPD spectrum of BWO does not exhibit a distinct characteristic peak due to the scarcity of rich oxygen defects and unsaturated W atoms. These findings suggest that Ni-induced the coordination-unsaturated W atoms can facilitate chemisorption of O_2 , resulting in the formation of activated oxygen species on the surface. This is expected to play an important role in the subsequent oxygen transfer process.

We thank the reviewer again for this important suggestion, which greatly improves our work.

Figure R4 Time-dependent in situ DRIFT spectra of 1.8 Ni/BWO (a) and BWO (b) in the dark under O_2 atmosphere. O_2 -TPD spectra of the prepared samples (c).

r1. Nayak, S., McPherson, I. J. & Vincent, K. A. Adsorbed Intermediates in Oxygen Reduction on Platinum Nanoparticles Observed by In Situ IR Spectroscopy. *Angew. Chem. Int. Ed.* **57**, 12855–12858 (2018).

r2. Wu, Q. et al. Unveiling the dynamic active site of defective carbon-based electrocatalysts for hydrogen peroxide production. *Nat. Commun.* **14**, 1-11 (2023).

r3. Zhang, X. et al. The promoting effect of H_2O on rod-like $MnCeO_x$ derived from MOFs for toluene oxidation: A combined experimental and theoretical investigation. *Appl. Catal. B Environ.* **297**, 120393 (2021).

Corresponding Revision: The description for the in situ DRIFT spectra has been revised as “In situ Diffuse Reflectance Infrared Fourier Transform (DRIFT) spectroscopy was applied to trace the adsorption process of O_2 and TL molecules. Figure 5a shows two absorption peaks appeared at 1476 cm^{-1} and 1290 cm^{-1} ,

corresponding to the vibrational peaks of adsorbed oxygen and the vibration model of superoxide O-O in surface-coordinated oxygen complexes, respectively.^{41, 42} In comparison, BWO exhibits a weaker signal (Figure 5b), suggesting that the unique CAUs of 1.8 Ni/BWO enhance the adsorption of O₂. ” Page line.

The sentences “O₂-TPD spectra (Figure 5c) also shows that 1.8 Ni/BWO has a desorption peak at around 370 °C, which is related to the chemical adsorbed oxygen species, corresponding to the O₂⁻ and O₂²⁻ adsorbed on unsaturated W atoms.⁴³ Ni-induced the coordination-unsaturated W atoms can facilitate chemisorption of O₂, resulting in the formation of activated oxygen species on the surface. This is expected to play an important role in the subsequent oxygen transfer process.” have been added to the revised manuscript.

The sentence “Moreover, the vibrational signal of adsorbed O₂ gradually becomes weaker, demonstrating the consumption of O₂ during the reaction.” has been revised as “Moreover, the vibrational signal of chemical adsorbed oxygen complexes (1290 cm⁻¹) gradually becomes weaker, demonstrating that there is a unique oxygen transfer process”.

Figure R4 (a) and (b) have been added to the revised manuscript as Figure 5 (a) and (b).

Figure R4 (c) has been added to the revised manuscript as Figure 5 (c).

The sentence “O₂ temperature programmed desorption (O₂-TPD) spectra were test on a AutoChem II (Micromeritics Instrument Corp.)” has been added to the **Methods**.

Reference r1-3 have been added to the revised manuscript as Reference 41-43 and the order of references has been updated.

Figure 5 (f, g and h) have been moved to the Supplementary information as Figure S25 (a, b and c).

5. The authors also highlights that “the O atoms from the broken O₂ molecule” can be used for the formation of BD and H₂O, however, the form of O atoms involved in the reaction is unclear (surface lattice O atoms or just O₂?). Additional experiments to support the hypothesis of the reaction pathway should be included and the clarity of the

mechanism can be improved.

Response: Thanks for your valuable comment. We have included additional O₂-TPD tests and ¹⁸O₂ isotope-labeling experiment. The O₂-TPD result (Figure R5) shows that 1.8 Ni/BWO has a strong ability to adsorb O₂, forming chemically adsorbed oxygen species (O₂⁻ and O₂²⁻). The chemical adsorbed oxygen species are gradually consumed under light irradiation, as traced by the in situ DRIFT spectra at 1290 cm⁻¹ (Figure R4). This indicates that these chemically adsorbed oxygen species are involved in the reaction and establish a new surface oxygen transfer process. These species are formed by the coordination activation of O₂ on unsaturated W. The ¹⁸O₂ isotope-labeling experiment demonstrates that 80% of the BD molecules are labeled by ¹⁸O when using ¹⁸O₂ as the oxygen source. This suggests that the O atoms in most benzaldehyde come from O₂, while the remainder come from the chemical adsorbed oxygen species on unsaturated W atoms when the sample is exposed to air, consistent with the results of in situ DRIFT spectra and O₂-TPD. Some O₂ molecules are reduced to •O₂⁻ radicals by photogenerated electrons, while others are activated to chemically adsorbed oxygen species on unsaturated W. Thus, the forms of O atoms involved in the oxygenation of toluene are •O₂⁻ radicals or chemical adsorbed oxygen species on unsaturated W. These chemical adsorbed oxygen species on unsaturated W would establish a new oxygen transfer process (Figure 6a).

In addition, we found that the lattice oxygen primarily originates from the oxygen atoms coordinated on the surface of tungsten (W) atoms, as proposed in the mechanism. A comparison was made between the O-O and W-O bond lengths of O₂ adsorbed on the W surface and lattice oxygen atoms coordinated with W, revealing that the lattice O is in a different chemical state from that of O in O₂. The lattice O-O bond length is 2.802 Å and the W-O bond length is 1.765 Å (Figure R5a). The adsorbed O₂, on the other hand, has an O-O bond length of 1.499 Å and a W-O bond length of 1.925 Å (Figure R5b). This indicates that the interaction between lattice O and W is much stronger than the W-O interaction in adsorbed oxygen species, making the migration process of lattice O may be more difficult. The activation energy for lattice O (6-coordinated O with W) to migrate to a 5-coordinated Ni atom was calculated to be 2.44 eV (with a reaction

energy of 1.91 eV). (Figure R5c) This is 2.1 eV higher than the oxygen migration barrier in adsorbed oxygen species. Therefore, the ^{16}O product in isotope experiments is very likely derived from the pre-adsorbed O_2 in the air. The lattice O involved mechanism is unlikely to occur in our system. Thanks again for this very valuable suggestion. The clarity of the mechanism has significantly been improved, especially regarding the adsorption of O_2 and the transfer of surface oxygen, after addressing these questions.

Corresponding Revision: The adsorption of O_2 and the new surface oxygen transfer process induced by the chemical adsorbed O species on unsaturated W is supplemented by O_2 -TPD test and $^{18}\text{O}_2$ isotope-labeling experiment.

The sentences “On the other hand, as the distance between two O atoms is stretched by W atoms in Figure 6a TS1, the O_2 molecule is eventually broken (model III). Interestingly, the O atoms from the broken O_2 molecule in model III can easily deliver to Ni atoms (TS2).” have been revised as “On the other hand, as the distance between two O atoms is stretched by W atoms in Figure 6a TS1, the chemical adsorbed oxygen species are eventually broken (model III). Interestingly, the O atoms from the broken chemical adsorbed oxygen species in model III can easily deliver to Ni atoms (TS2).”

The sentence “Considering the difficulty for lattice oxygen on tungsten (W) to participate in reactions (Figure S23), the remaining O atoms are considered to be the chemical adsorbed oxygen species on unsaturated W atoms when the sample is exposed to air, consistent with the results of in situ DRIFTS and O_2 -TPD.” has been added to the revised manuscript.

Figure R5 have been added to the Supplementary information as Figure S23. The order of the images has been updated.

The description “A comparison was made between the O-O and W-O bond lengths of O_2 adsorbed on the W surface and lattice oxygen atoms coordinated with W, revealing that the lattice O is in a different chemical state from that of O in O_2 . The lattice O-O bond length is 2.802 Å and the W-O bond length is 1.765 Å (Figure S23a). The adsorbed O_2 , on the other hand, has an O-O bond length of 1.499 Å and a W-O bond length of 1.925 Å (Figure S23b). This indicates that the interaction between lattice

O and W is much stronger than the W-O interaction in adsorbed oxygen species, making the migration process of lattice O may be more difficult. The activation energy for lattice O (6-coordinated O with W) to migrate to a 5-coordinated Ni atom was calculated to be 2.44 eV (with a reaction energy of 1.91 eV). (Figure S23c) This is 2.1 eV higher than the oxygen migration barrier in adsorbed oxygen species. Therefore, the ^{16}O product in isotope experiments is very likely derived from the pre-adsorbed O_2 in the air. The lattice O involved mechanism is unlikely to occur in our system.” has been added to the Supplementary information.

Figure R5 Optimized structures of chemisorbed O_2 (a) and lattice O (b) on Ni/BWO. The lattice O transfer mechanism with the corresponding energies (c).

6. *Can the authors comment on how the mass fractions of Ni (0.9, 1.8, 2.6) influence the photocatalytic performance?*

Response: Thanks for your valuable comment. The number of defects and active sites induced varies with different mass fractions of Ni. XPS and EPR data indicate that when the mass fraction of Ni is lower (0.9%), fewer active centers are produced, resulting in an insignificant improvement in toluene oxidation performance. Conversely, an excessively high mass fraction of Ni (2.6%) does not result in a greater number of active sites. When the mass fraction of Ni is 1.8%, it produces the highest concentration for W^{5+} and FLPs, which is favorable for the adsorption and activation of reactants. Surface defects also facilitate the separation of photogenerated electron-hole pairs, thereby enhancing photocatalytic performance.

Corresponding Revision: The sentence “The reason is that the number of defects and active sites induced varies with different mass fractions of Ni. When the mass fraction of Ni is 1.8%, it produces the highest concentration for W^{5+} and FLPs (Figure 3), which is favorable for the adsorption and activation of reactants. Surface defects also facilitate the separation of photogenerated electron-hole pairs (Figure S25),⁵¹ thereby enhancing photocatalytic performance.” has been added to the revised manuscript.

Reviewer #2 (Remarks to the Author):

This manuscript focuses on the photocatalytic selective oxidation of toluene via the creation of nickel-mediated cascaded active units as photocatalytic active centers. These cascaded active units include unsaturated W atoms, oxygen vacancies and Bi/O frustrated Lewis pairs (FLPs), which not only enable the efficient activation of C-H bonds and O₂, but also facilitate the transport of the active O species, thus promoting a very efficient catalytic process. Overall, the manuscript is well organized in terms of materials characterization and mechanistic studies. It is an interesting paper with novel insights for future catalyst design. The following minor issues should be addressed before acceptance.

1. The photocatalytic reaction is irradiated using a 300 W Xenon lamp. Exposure to light generates heat, how is the temperature of the catalytic system maintained? What is the effect of temperature on reaction performance?

Response: Thanks for your valuable comment. Our photocatalytic system employs an electric fan for cooling to suppress the temperature rise of the catalytic system. Generally, weak thermal effects are difficult to avoid completely. We test the performance of the prepared 1.8 Ni/BWO at different temperatures in the dark to study the thermal effects. The result in Figure R6 shows that no benzaldehyde is produced in the dark, regardless of temperature. It confirms that the oxidation of toluene is driven by light. The thermal effect in this photocatalytic system is too weak to influence the reaction performance.

Corresponding Revision: The sentence “No benzaldehyde is produced in the dark, regardless of temperature (Figure S20a). It confirms that the oxidation of toluene is driven by light.” has been added to the revised manuscript. In addition, Figure R6 has been added to the Supplementary information as Figure S20a.

Figure R6 The yield of benzaldehyde at different temperature in the dark over 1.8 Ni/BWO.

2. *The selectivity of the products and the turnover frequency (TOF) should be evaluated.*

Response: Thanks for your valuable comment. As shown in Figure R1a, the selectivity of the product in different reaction times was calculated and we conducted the reproducibility studies and added the error bars. When 1.8Ni/BWO is used for photocatalytic toluene oxidation, benzyl alcohol could be detected with a selectivity of 9% in the primary stage (reaction time 2h). As the reaction time increases, benzyl alcohol is further converted to benzaldehyde with a selectivity of over 95% and a small amount of benzaldehyde is gradually oxidized to benzoic acid. Interestingly, the oxidation of benzaldehyde to benzoic acid is more obvious on BWO (Figure R1b). This may be related to the fact that the presence of CAUs promotes the conversion of the oxidation product to BD. The selectivity of benzaldehyde was always maintained above 90%. Thus, the catalyst with a Ni mass fraction of 1.8% exhibits excellent toluene conversion rates and high selectivity towards benzaldehyde (Abstract, line 27).

Considering toluene as the reaction solvent, the toluene conversion is very low after 12 hours of reaction, which does not reflect the product distribution of toluene

under high conversion conditions. In order to observe the product distribution of toluene at a high conversion, we used acetonitrile as the solvent and added 0.1 mmol of toluene as the reactant to conduct photocatalytic performance tests. The result is shown in Figure R1c and d. The main product is still BD at high TL conversion. After the complete conversion of TL, BD is further oxidized to BAC. This result suggests that the CAUs promotes the conversion of the oxidation product to BD and the further oxidation of BD to BAC is suppressed in the presence of TL.

We calculate the TOF according to the formulas in previous study using the moles of the catalyst^{r4}.

TOF = moles of consumed toluene / (moles of catalyst × reaction time).

The calculated TOF is $2.2 \pm 0.2 \text{ h}^{-1}$.

r4. Nikoloudakis, E. et al. Dye-Sensitized Photoelectrosynthesis Cells for Benzyl Alcohol Oxidation Using a Zinc Porphyrin Sensitizer and TEMPO Catalyst. *ACS Catal.* **11**, 12075–12086 (2021).

Corresponding Revision: The sentences “A small number of benzyl alcohol (BA) could be detected at the primary stage. As the reaction time increases, these few BA molecules are further converted to BD. The selectivity of benzaldehyde has consistently been maintained at over 90%, even under the conditions of high toluene conversion (Figure S19a and b). Upon complete conversion of TL, BD is further oxidized to BAC.” have been added to the revised manuscript. (page line)

“possibly due to the presence of CAUs promoting the conversion of the oxidation product to BD and suppressing the further oxidation of BD to BAC in the presence of TL.” has been added to the revised manuscript. (page line)

“with a turnover frequency (TOF) about 2.2 h^{-1} ” has been added to the revised manuscript. (page line)

The sentences “Turnover frequency (TOF) is calculated according to the formulas in previous study using the moles of the catalyst.⁵²

TOF = moles of consumed toluene / (moles of catalyst × reaction time).” have been added to the Supplementary information.

Figure R2 (a) and (b) have been added to the revised manuscript as Figure 4 (b)

and (c).

Figure R2 (c) and (d) have been added to the Supplementary information as Figure S19 (a) and (b).

Reference r4 have been added to the Supplementary information as Reference 3.

Figure R1 The time-dependent process of the oxidation of TL over 1.8Ni/BWO (a) and BWO (b), error bars are mean \pm SD based on 3 repeat experiments. Gas phase (c) analysis for different reaction time from our online GC setup and the corresponding conversion of TL and selectivity of BD (d); reaction condition: 10 mg 1.8 Ni/BWO, 0.1 mmol TL, 1.5 mL acetonitrile, O₂.

3. In the simulation of the catalytic mechanism, have the authors considered the activation of O₂ by the unsaturated Bi atom, while the O atom and the unsaturated W atom serve as the activation sites for toluene? Can the O and W atoms also serve as FLP sites for toluene activation?

Response: Thank you for your professional suggestion. During our previous computation process, we had already considered using unsaturated bismuth (Bi) or Ni dopant as a site for O₂ activation. However, according to our computational results, the adsorption of O₂ on Bi atoms and Ni dopant is weaker than that on tungsten (W), as the calculated adsorption energies of O₂ on unsaturated Bi, Ni, and W sites are 0.31, 0.14,

and -0.10 eV, respectively (Table R1). Therefore, O₂ molecules are more likely to adsorb on unsaturated W sites and then be activated.

Table R1 Calculated adsorption energy for O₂ on unsaturated Bi, Ni and W sites.

Site	Adsorption energy of O ₂ (eV)
Bi	0.31
Ni	0.14
W	-0.10

Corresponding Revision: The sentence “As the calculated adsorption energies of O₂ on unsaturated Bi, Ni, and W sites are 0.31, 0.14 and -0.10 eV, (Table S8) respectively,” has been added to the revised manuscript.

Table R1 has been added to the Supplementary information as Table S8.

4. DFT calculations indicate that by doping with Ni, the activation barrier for C-H bond activation can be as low as 0.3 eV, while the spillover of activated O atoms to the FLP site becomes the rate-determining step with a high activation energy of 0.59 eV. Have the authors considered reducing the barrier for the spillover of O atoms (e.g. by designing a type of NiM bimetallic catalyst) to achieve more efficient oxidation of toluene?

Response: Thank you for your suggestion. We did not consider the possibility of further optimizing the catalytic performance based on our current work. From the conclusions of our current work, your proposed plan is indeed valuable and innovative. We plan to further optimize the activity of our catalyst through doping, surface modification, and other modification strategies in our future work, and also to re-examine the mechanisms we have proposed.

Reviewer #3 (Remarks to the Author):

Shi et al. report the nickel-mediated cascaded active units in monolayer Ni/Bi₂WO₆ for photocatalytic selective toluene oxidation. The cascaded active units (CAU) consist of unsaturated W atoms and Bi/O frustrated Lewis pairs (FLPs). A very interesting finding from the DFT and experimental results is that the CAU can promote both the activation of C–H bonds and O₂ and the transportation of active O, thereby establishing a very efficient C–H bond oxidation process. In my opinion, the creation of cascaded active units is innovative and this work was well organized. I am very impressed with this work, which provides a significant new contribution that highlights the strategy of creating CAUs with synergistic effects for photocatalytic toluene oxidation. The insights gained from this work can help accelerate the development of related photocatalysts. Therefore, I recommend the publication of this manuscript after addressing some minor issues as shown below:

1. Figure 2g shows that some O_{Vs} are present in the area without Ni dopants. What is the reason for the formation of these O_{Vs}? Do these O_{Vs} promote the photocatalytic performance.

Response: Thanks for your good questions. Some O_{Vs} are present in the area without Ni dopants. These may be attributed to the spontaneous formation of O_{Vs} on Bi₂WO₆ monolayer nanosheets. These O_{Vs} can also enhance the photocatalytic performance. However, there are no CAUs in the vicinity of these O_{Vs}. According to the catalytic performance of BWO-O_V, and DFT calculation results of BWO, it can also be proved that the promotional effect of these isolated oxygen O_{Vs} on the performance is not significant compared to the CAUs.

Corresponding Revision: The sentences “Some O_{Vs} are present in the area without Ni dopants. These may be attributed to the spontaneous formation of O_{Vs} on Bi₂WO₆ monolayer nanosheets.” have been added to the revised manuscript.

2. It is found that 1.8 Ni/BWO has more surface hydroxyl group species than other samples from the XPS results. The influence of surface hydroxyl group species on the

photocatalytic performance should be discussed.

Response: Thanks for your good suggestion! These surface –OH groups are induced by Ni dopants. The results of Figure S25b and Table S8 show that transient surface –OH groups and Bi atoms are also suitable FLPs for the activation of C-H bonds. Thus, more surface hydroxyl groups on 1.8 Ni/BWO also indicate the highest level of FLPs on 1.8 Ni/BWO, which promotes the photocatalytic performance.

Corresponding Reversion: The sentences “More surface hydroxyl groups on 1.8 Ni/BWO (XPS results) also indicate the highest levels of FLPs on 1.8 Ni/BWO, which promotes the activation and dehydrogenation of the C-H bond.” have been added to the revised manuscript.

3. What is the procedure for the regeneration of the produced catalysts during the cycle experiments?

Response: Thanks for your suggestion, which is very valuable for our work. The regeneration of produced catalysts is conducted by a simple method. The methods are displayed as follows: The used catalyst is cleaned with methanol for 3 times and then collected by centrifugation. The collected catalyst as the regenerated catalyst is dried in the drying oven at 60 °C.

Corresponding Revision: The details “The used catalyst is cleaned with methanol for 3 times and collected after drying in 60 °C for 6 h to conduct the cycle experiments.” has been added to the **Methods**.

4. The author focuses on analyzing the significance of Ni-mediated unsaturated W for the C-H activation of toluene, but seldom discusses the activation process of the phenyl group's C-H bond. The entire process of moderate oxidization of toluene to benzaldehyde also involves the activation of the C-H bond in the phenyl group. The author should supplement whether the unique active center has a similar effect on the activation of C-H bond in the phenyl group as it does on the activation of the C-H bond in toluene.

Response: Thank you very much for your insightful comments. In our original

manuscript, we have already elaborated that our FLP sites are very stable. This is because after the activation of the C-H bond of toluene ($-\text{CH}_3$), the generated hydrogen (H) atom will spill over to the oxygen atom coordinated with the Ni atom, forming an OH group. Subsequently, our calculation results of Figure S25b and Table S8 revealed that the Ni-OH state can also serve as a stable FLP site to coordinate with the benzyl group ($-\text{CH}_2$). The surface $-\text{OH}$ combines with the H atom in the C-H bond to form a desorbed H_2O molecule, leaving a low-coordinated Ni atom and the $^*\text{C}_7\text{H}_6$ intermediate. The activated O atoms on the unsaturated W atoms would transfer to the Ni dopant to oxygenate the $^*\text{C}_7\text{H}_6$ intermediate to BD and replenish the O consumption of dehydration to start the next cycle. Thus, the unique active center (FLP) has a similar effect on the activation of the C-H bond in the phenyl group as it does on the activation of the C-H bond in toluene.

Reviewer #4 (Remarks to the Author):

Lin et al reported a series of Ni-doped monolayer Bi₂WO₆ nanosheets (Ni/BWO) with varying Ni mass fractions for the photocatalytic selective oxidation of toluene. The composition of the Ni/BWO was analyzed by ICP-MS, TEM and AFM. In the photocatalytic oxidation of toluene, a high selectivity of benzaldehyde was obtained. However, the performance of the current manuscript is not innovative enough. So, the work needs major revisions before it can be considered for publication in Nature Communication.

1. The introduction of the second metal changes the electron configuration around the original metal. (Angew Chem Int Ed 2018, 57, 9660.) The exact role of Ni should be carefully studied. Is Ni the catalytic active site or is Ni replacing W affecting the electron configuration around W, and the catalytic active site is still W?

Response: Thank you very much for your professional insights. In our catalyst system, Ni plays an important role, but it does not participate in the reaction directly. The most important role of Ni dopants is to induce cascaded active units consisting of unsaturated W atoms, O_vs and Bi/O frustrated Lewis pairs (FLPs). Ni replaces W atoms in the subsurface, so the Ni dopants are not the catalytic active site. Loading Ni on the surface of the Ni/BWO-surface sample without O_v and unsaturated W, as well as FLPs, shows negligible improvement for the conversion of TL, indicating that the actual active sites are O or other surrounding atoms such as W or Bi rather than Ni.

Indeed, as stated in the literature¹⁵, doping can lead to a redistribution of charge around neighboring atoms, thereby altering their electronic structure. The Bi atoms around Ni dopants have different electron densities from those on pristine BWO (Figure S12), implying higher activity. This has been described in the original manuscript. However, we found that the closest distances between the unsaturated W and the Ni atom in CAUs are as large as 5.435 Å (W-Ni). Thus, the effect of Ni on the electron configuration around unsaturated W is minor.

During our previous computation process, we had also considered using unsaturated bismuth (Bi) or Ni dopant as a site for O₂ activation. However, according

to our computational results, the adsorption of O₂ on Bi atoms and Ni dopant is weaker than that on W, as the calculated adsorption energies of O₂ on unsaturated Bi, Ni, and W sites are 0.31, 0.14 and -0.10 eV, (Table R1) respectively. Therefore, unsaturated W sites are more likely to adsorb O₂ molecules.

Furthermore, as shown in Figure R7, our previous calculations on toluene adsorption included the consideration of potential surrounding sites (such as Ni, Bi, etc.). The results showed that the current FLP adsorption system for toluene has the lowest energy. Therefore, it can be concluded that FLP are effective in adsorbing the C-H bonds.

Thus, Ni changes the electron density of Bi in FLPs, making it more active, but has a minor effect on the electron configuration around W. The FLP site (formed by Ni-O and Bi) is the adsorption and activation site for toluene, while the unsaturated W site is the adsorption and activation site for O₂.

We appreciate the professional questions raised by the reviewers. Also, we regret that at present, we are unable to provide direct experimental evidence to demonstrate whether W is the active site for C-H bond activation. This is an interesting topic that we will focus on in future work.

r5. Wang, X. L. et al. Exploring the Performance Improvement of the Oxygen Evolution Reaction in a Stable Bimetal–Organic Framework System. *Angew. Chem. Int. Ed.* **57**, 9660–9664 (2018).

Table R1 Calculated adsorption energies for O₂ on unsaturated Bi, Ni and W sites.

Site	Adsorption energy of O ₂ (eV)
Bi	0.31
Ni	0.14
W	-0.10

Figure R7 The optimized adsorption structures of the C-H bond in toluene on different surface sites.

Corresponding Revision: The sentence “As the calculated adsorption energies of O₂ on unsaturated Bi, Ni, and W sites are 0.31, 0.14 and -0.10 eV, (Table S8) respectively,” has been added to the revised manuscript

The sentence “Furthermore, since doping can lead to a redistribution of charge around neighboring atoms,⁵² we considered the influence of potential surrounding sites (such as Ni and Bi site) and found that the current FLP adsorption system for toluene has the lowest energy (Figure S26a).” has been added to the revised manuscript.

Figure R7 has been added to the Supplementary information as Figure S26a.

Reference r5 have been added to the Revised manuscript as Reference 52.

Table R1 has been added to the Supplementary information as Table S8.

2. Not only comparing with the BWO-Ov, BiOCl and so on, summarizing a list including the performance of other reported catalysts for toluene oxidation is necessary.

Response: Thanks for your valuable comment. As shown in Table S6, we have summarized a list including the performance of other reported catalysts for toluene oxidation. We also included some of the most recent representative papers in Table R2 for comparison. It can be observed that 1.8Ni/BWO shows better performance than most of the other photocatalysts, indicating that the construction of CAUs via in situ lattice substitution of Ni dopant is a highly efficient strategy to improve the photocatalytic oxidation of TL.

Table R2 Previous studies for the photocatalytic TL oxidation.

Catalyst	Reaction condition	Rate ($\mu\text{mol g}^{-1} \text{h}^{-1}$)	Ref.
1.8 Ni/BWO	$\lambda \geq 400 \text{ nm}$, sample (10 mg), toluene (2 mL),	4560	This work
Fe-UiO-66	$\lambda \geq 380 \text{ nm}$, sample (10 mg), toluene (5 μL)	1295	r6
0.01BiOCl/TiO ₂	$\lambda = 365 \text{ nm}$, sample (25 mg), toluene (1 mmol)	2000	r7
Y ₁ /TiO ₂	sample (50 mg), toluene (0.5 mmol)	850	r8
Cs ₄ ZnSb ₂ C ₁₁	$\lambda \geq 400 \text{ nm}$, sample (10 mg), toluene (1 mL)	1893	r9
Cs ₂ AgBiBr ₆ /CN	sample (100 mg), toluene (5 mL)	2630	r10

r6. Xu, C. et al. Turning on Visible-Light Photocatalytic C-H Oxidation over Metal-Organic Frameworks by Introducing Metal-to-Cluster Charge Transfer. *J. Am. Chem. Soc.* **141**, 19110–19117 (2019).

r7. Wang, H. et al. Achieving High Selectivity in Photocatalytic Oxidation of Toluene on Amorphous BiOCl Nanosheets Coupled with TiO₂. *J. Am. Chem. Soc.* **145**, 16852–16861 (2023).

r8. Xue, Z. et al. Efficient Benzylic C-H Bond Activation over Single-Atom Yttrium Supported on TiO₂ via Facilitated Molecular Oxygen and Surface Lattice Oxygen Activation. *ACS Catal.* **14**, 249–261 (2024).

r9. Mai, H. et al. Synthesis of Layered Lead-Free Perovskite Nanocrystals with Precise Size and Shape Control and Their Photocatalytic Activity. *J. Am. Chem. Soc.* **145**, 17337–17350 (2023).

r10. Song, J. et al. In situ growth of lead-free perovskite Cs₂AgBiBr₆ on a flexible ultrathin carbon nitride sheet for highly efficient photocatalytic benzylic C(sp³)-H bond activation. *Chem. Eng. J.* **453**, 139748 (2023).

Corresponding Revision: Reference r6-10 have been added to the Supplementary information as Reference 14-18.

Table S6 has been revised to Table S7 as follow:

Table S7 Previous studies for the photocatalytic TL oxidation.

Catalyst	Reaction condition	Rate ($\mu\text{mol g}^{-1} \text{h}^{-1}$)	Ref.
----------	--------------------	---	------

1.8 Ni/BWO	$\lambda \geq 400$ nm, sample (10 mg), TL (2 mL),	4560	This work
Bi ₂ W _{0.3} Mo _{0.7} O ₆	$\lambda \geq 420$ nm, sample (15 mg), TL (1 mL)	1663	5
p-BWO	$\lambda \geq 400$ nm, sample (250 mg), TL (8 mmol)	3453	6
p-BWO	$\lambda \geq 400$ nm, sample (250 mg), TL (10 mmol),	4388	6
Fe(0.26)-BWO	$\lambda \geq 400$ nm, sample (20 mg), TL (10 mmol)	1304	7
Pd/Bi ₂ WO ₆	$\lambda \geq 400$ nm, sample (50 mg), TL (1 mL)	1267	8
Flower-like Bi ₂ WO ₆	$\lambda \geq 400$ nm, sample (50 mg), TL (8 mmol)	464	9
HMNRs	No cut-off filter, sample (20 mg), TL (3 mL)	2254	10
Cs ₃ Sb ₂ Br ₉	$\lambda \geq 420$ nm, sample (10 mg), TL (5 mL)	2520	11
Cs ₃ Bi ₂ Br ₉ (1.7)/d- BiOBr	$\lambda \geq 400$ nm, sample (10 mg), TL (5 mL)	7240	12
10wt%Cs ₃ Bi ₂ Br ₉ /SBA-15	$\lambda \geq 420$ nm, sample (10 mg), TL (5 mL)	12600	13
Fe-UiO-66	$\lambda \geq 380$ nm, sample (10 mg), TL (5 μ L)	1295	14
0.01BiOCl/TiO ₂	$\lambda = 365$ nm, sample (25 mg), TL (1 mmol)	2000	15
Y ₁ /TiO ₂	sample (50 mg), TL (0.5 mmol)	850	16
Cs ₄ ZnSb ₂ C ₁₁	$\lambda \geq 400$ nm, sample (10 mg), TL (1 mL)	1893	17
Cs ₂ AgBiBr ₆ /CN	sample (100 mg), TL (5 mL)	2630	18

3. Please supplement the photocatalytic oxidation properties of toluene derivatives with Ni/BWO as the photocatalyst.

Response: Thanks for your valuable comment! The experiments for the photocatalytic oxidation of toluene derivatives are conducted. Table R2 show that the toluene derivatives with electron-donating group (-NH₂) are easier to convert to the product than those with electron-withdrawing groups (-F, -Cl, -NO₂). The selectivity of the corresponding benzaldehyde derivatives is reduced due to the removal of other substitutions. The result shows that 1.8 Ni/BWO exhibits good conversion of other toluene derivatives.

Corresponding Revision: The sentence “This photocatalyst also shows good conversion for the oxidation of other toluene derivatives. (Table S6)” has been added to the revised manuscript.

The sentences “The experiments for the photocatalytic oxidation of toluene derivatives are conducted. Table S6 show that the toluene derivatives with electron-donating group (-NH₂) are easier to convert to the product than those with electron-withdrawing groups (-F, -Cl, -NO₂). The selectivity of the corresponding benzaldehyde derivatives is reduced due to the removal of other substitutions. The result shows that 1.8 Ni/BWO exhibits good conversion of other toluene derivatives.” Has been added to the Supplementary information.

Table R3 has been added to the as Table S6.

Table R3 Photocatalytic oxidation of toluene derivatives.

Reactant	Product	Con. (%)	Sel. (%)
		48	99
		36	62
		56	51
		37	63
		42	68
		29	71

Reaction condition: 10 mg 1.8 Ni/BWO, 0.1 mmol toluene derivatives, 1.5 mL acetonitrile, O₂, reaction time 5h.

4. “BA is further converted to BD and a small amount of BD is oxidized to benzoic acid (BAC)” in line 283. However, toluene can also be reported to oxidize directly to benzaldehyde (10.1021/jacs.9b09954). Please give the evidence that BA is converted to BD, not TL directly oxidized to BD.

Response: Thanks for your valuable comment. We agree with your question regarding the lack of precision in our description. Our study reveals the existence of two reaction routes in this photocatalytic system. The first route involves CAUs, which directly convert TL to BD (as shown in Figure 6e). The second route is a photo-induced free radical process, which is difficult to control. Figure R1 and in situ DRIFT spectra indicate that only a small amount of benzyl alcohol is produced in the initial stage of the reaction and is further converted to BD over time. A possible mechanism for the oxidation of toluene is provided based on semiconductor energy band theory and photo-induced active radical, as shown in Figure R8. TL can be directly converted to BD, while the formation of BA is a minor process and BA can be further oxidized to BD.^{r6,}
6

r6. Xu, C. et al. Turning on Visible-Light Photocatalytic C-H Oxidation over Metal-Organic Frameworks by Introducing Metal-to-Cluster Charge Transfer. *J. Am. Chem. Soc.* **141**, 19110–19117 (2019).

Corresponding Revision: The sentence “BA is further converted to BD and a small amount of BD is oxidized to benzoic acid (BAC)” has been replaced as “A small number of benzyl alcohol (BA) could be detected at the primary stage. As the reaction time increases, these few BA molecules are further converted to BD. The selectivity of benzaldehyde has consistently been maintained at over 90%, even under the conditions of high toluene conversion (Figure S19a and b). Upon complete conversion of TL, BD is further oxidized to BAC.”

Figure S27 has been replaced by Figure R8.

Reference r6 has been referenced to provide a more reasonable photo-induced free radical process.

Figure R1 The time-dependent process of the oxidation of TL over 1.8Ni/BWO (a) and BWO (b), error bars are mean \pm SD based on 3 repeat experiments. Gas phase (c) analysis for different reaction time from our online GC setup and the corresponding conversion of TL and selectivity of BD (d); reaction condition: 10 mg 1.8 Ni/BWO, 0.1 mmol TL, 1.5 mL acetonitrile, O₂.

Figure R8 Possible mechanism for the oxidation of toluene based on semiconductor energy band theory and photo-induced active radical.

5. Pay attention to writing norms. (C_{sp}³-H, in line 319 and 332)

Response: Thanks for your careful check. C_{sp}³-H (in line 319 and 332) has been revised as C-H.

Corresponding Revision: “C_{sp}³-H” has been revised as “C-H”.

REVIEWERS' COMMENTS

Reviewer #1 (Remarks to the Author):

Comments to the authors:

The concerns that we raised are all properly addressed. Now I'm sure that this revised manuscript can be accepted for publication.

Reviewer #2 (Remarks to the Author):

The authors have well addressed my comments. Acceptance in the revised MS is recommended.

Reviewer #3 (Remarks to the Author):

Authors have carefully revised manuscript according to reviewers' comments. It should be accepted.

Reviewer #4 (Remarks to the Author):

The authors have addressed my questions and concerns. And I think I am fine with the acceptance for the manuscript.